# Provably Efficient and Practical Self-Play for Better LLM Alignment

## Abstract

Reinforcement Learning with Human Feedback (RLHF) has gained significant attention for aligning AI behavior with human preferences. Self-play style RLHF has shown strong advantages, as highlighted by many studies. However, current self-play style RLHF approaches face several limitations, including the lack of provable sample efficiency, absence of active exploration, and limited diversity in training data. To address these challenges, we propose a novel RLHF framework that balances exploration and exploitation while providing theoretical guarantees. We introduce Two-Agent Nash Policy Optimization (TANPO) as an equivalent and easy-to-implement two-agent algorithm building on this framework. In TANPO, the two players are trained using different loss functions to ensure more diverse and informative data collection. We also propose Single-Agent Diversity-driven Optimization (SADPO), a single-agent approximation of TANPO, supported by both theoretical analysis and empirical evidence. Our theoretical analysis shows that our theoretical algorithm framework enjoys sublinear regret under general function approximation and mild structural conditions, with a detailed analysis provided for the linear case. Empirically, we implement TANPO and SADPO using Zephyr-7B-SFT as our base model, outperforming several baselines across multiple evaluation benchmarks, such as AlpacaEval 2.0, MT-Bench and various standard academic benchmarks. Our experiments also show that TANPO improves performance on AlpacaEval 2.0 over extended training epochs, demonstrating its ability to consistently improve and reduce overfitting.

## 1 Introduction

Large Language Models (LLMs) have shown significant proficiency in understanding and generating natural language. Reinforcement Learning with Human Feedback (RLHF) is key to improving LLMs by directly integrating human feedback into their training (Christiano et al., 2017; Ziegler et al., 2019). This process typically involves training the model using reinforcement learning techniques to maximize the reward from human-labeled data (Ouyang et al., 2022). Many RLHF algorithms treat this setup as a contextual bandit problem, where the prompt corresponds to the state, the generated response represents the action, and the feedback received acts as the reward.

Another approach, in addition to modeling RLHF as a contextual bandit problem, is to utilize self-play methods. Self-play is a technique in which one or more agents learn by competing against themselves, allowing them to iteratively improve their strategies by evaluating and adapting to their own responses. This approach has proven to be a powerful method in various fields, including traditional reinforcement learning such as AlphaGo Zero (Silver et al., 2017) and generative models such as GANs (Goodfellow et al., 2020). In the context of RLHF, self-play algorithms typically involve one or multiple agents generating multiple responses for each prompt. These responses compete against each other through feedback provided either by human evaluators or AI annotators, which is then used to train the agents. Recently, a line of works (Rosset et al., 2024; Wu et al., 2024; Ye et al., 2024; Munos et al., 2023) have proposed a variety of self-play style RLHF algorithms. The goal of these algorithms is to find the Nash equilibrium strategy. These works have demonstrated the effectiveness of self-play algorithms in improving LLM performance.

Despite recent advances, there are several limitations in current self-play style RLHF approaches. *First*, there lacks theoretical guarantee on how these practical self-play algorithms can approximate Nash equilibrium, or there is a significant gap between theoretically guaranteed algorithms and practical implementations. *Second*, most existing algorithms lack active online exploration, which limits their ability to efficiently gather informative data during the learning process. Theoretically, active exploration can provide formal guarantees for learning efficiency (Xiong et al., 2023; Ye et al., 2024). Practically, it enhances model performance by ensuring that the training data remains diverse and novel, leading to better generalization and improved model performance (Zhang et al., 2024; Xie et al., 2024).

Therefore, a key research problem is: *Can we design an easy-to-implement and provably efficient self-play RLHF algorithm that approximates Nash equilibrium with active exploration?*

In this paper, we propose a new self-play RLHF algorithm. To effectively balance the trade-off between exploration and exploitation, the max-player aims to maximize the summation of **(i)** the expected Nash equilibrium value function and **(ii)** the negative estimation loss of that reward function. Similarly, the min-player seeks to maximize the summation of **(i)** the expected best response value function based on the max-player's strategy and **(ii)** the negative estimation loss of that reward function. We provide theoretical guarantee for this framework, showing that it achieves a sublinear regret under mild structural conditions.

We demonstrate that, under certain conditions, our algorithm framework is equivalent to an easy-to-implement algorithm. In this practical implementation, the max-player optimizes an MLE loss, while the min-player optimizes an MLE loss with a simple exploration bonus. The min-player's inclination towards exploration leads to more diverse and novel outputs, whereas the max-player's responses tend to align more closely with the reference policy. This dynamic contrast enables both players to engage with a broader range of information, ultimately improving their overall performance. Supported by both theoretical analysis and empirical evidence, we propose a single-agent algorithm that mimics the behavior of the two-agent algorithm.

**Contributions.** The main contributions of our work are as follows.

1. We introduce a theoretical two-player RLHF framework that effectively balances exploitation and exploration while providing a theoretical guarantee. Building on this framework, we propose a practical and easy-to-implement two-agent algorithm TANPO, where both players have simple and practical objectives. Additionally, we propose a single-agent algorithm SADPO that approximates the two-agent algorithm.

2. We prove that our theoretical algorithm achieves sublinear regret under general function approximation and mild structural conditions. We specify this result to a linear case and then provide a detailed regret analysis, showing our theoretical algorithm achieves a sublinear regret.

3. We implement our algorithms, along with several baselines, using Zephyr-7B-SFT (Tunstall et al., 2023) as the base model and the UltraFeedback dataset for prompts. Our algorithm outperforms several baseline methods across multiple evaluation benchmarks, including AlpacaEval 2.0, MT-Bench, PairRM win rate and various academic benchmarks. Additionally, we demonstrate that our algorithm continues to improve performance during a second epoch on the same dataset, highlighting its ability to achieve consistent gains and mitigate overfitting.

**Related Works.** We refer readers to Appendix A for a detailed discussion.

## 2 PRELIMINARIES

**RLHF pipeline.** RLHF leverages human preferences to guide the training of a language model. A common approach is the pairwise preference model, where feedback is provided by comparing two model-generated responses to the same prompt. We define an LLM as a policy $\pi(\cdot|\cdot)$ in policy class $\Pi$, where it takes a prompt $x$ and generates a response $a$ from distribution $\pi(\cdot|x)$. Given a prompt $x$ from the state space $\mathcal{X}$, a language model $\pi_\theta$ generates two candidate responses $a^1, a^2$ from the action space $\mathcal{A}$ according to its policy $\pi_\theta(\cdot|x)$. Human evaluators, or a reward model trained

to approximate human preferences, provide feedback in the form of a binary label $y \in \{0, 1\}$, indicating a preference for $a^1 \succ a^2$ when $y = 1$ or $a^2 \succ a^1$ when $y = 0$. This preference is modelled probabilistically using BT model (Bradley & Terry, 1952)

$$\mathbb{P}_r(y = 1|x, a^1, a^2) = \frac{\exp(r(x, a^1))}{\exp(r(x, a^1)) + \exp(r(x, a^2))} = \sigma(r(x, a^1) - r(x, a^2)). \tag{1}$$

Here, $r(x, a)$ is the human-provided score or a score predicted by a reward model that reflects the quality or suitability of response $a$ given the prompt $x$, and $\sigma(z) = 1/(1 + \exp(-z))$ denotes the sigmoid function.

In methods without a reward model, it has been shown that the preference loss can be expressed as a function of the policy. In preference optimization methods like DPO (Rafailov et al., 2024), the model is assumed to maximize a KL-regularized reward. Given a static dataset $\mathcal{D} = \{(x_i, a_i^+, a_i^-)\}_{i=1}^N$ of $N$ preference pairs, the parameterized reward model is learned by minimizing the following logistic regression loss

$$\mathcal{L}(r|\mathcal{D}) = -\mathbb{E}_{(x, a^+, a^-) \sim \mathcal{D}} \left[ \log \sigma(r(x, a^+) - r(x, a^-)) \right]. \tag{2}$$

**Two-Agent Zero-Sum Games.** In a two-agent zero-sum game, two players, termed the *max-player* and the *min-player*, engage in a competitive interaction where the gain of one player is exactly offset by the loss of the other. The game is characterized by a general value function $V(\pi, \mu)$, where $\pi$ and $\mu$ denote the mixed strategy probability distributions of the max-player and the min-player, respectively. In the Nash equilibrium, the max-player's strategy and the min-player's strategy are mutual best responses, meaning each is optimal given the strategy of the other (Nash et al., 1950). Formally, the max-player's strategy $\pi^*$ and the min-player's strategy $\mu^*$ solve the optimization problem given by

$$(\pi^*, \mu^*) = \arg \max_{\pi \in \Pi} \min_{\mu \in \Pi} V(\pi, \mu),$$

where $V(\pi, \mu)$ is a general function that captures the payoffs based on the strategies $\pi$ and $\mu$. The strategies $\pi^*$ and $\mu^*$ at this equilibrium are known as the Nash equilibrium strategies for the game.

**Performance Metric.** The goal of our learning algorithm is to find a policy $\pi$ for the max-player that is close enough to the Nash equilibrium. Consistent with the previous works (Ye et al., 2024; Liu et al., 2024a; Xie et al., 2020), we define the corresponding regret after $T$ episodes as

$$\text{Regret}(T) = \sum_{t=1}^{T} \left[ V(\pi^*, \mu^*) - V(\pi^t, \dagger) \right],$$

where $\pi^t$ is the policy used by the max-player in the $t$-th episode and $V(\pi^t, \dagger) := \min_{\mu \in \Pi} V(\pi^t, \mu)$. The target of sample efficient self-play style algorithm is to achieve a sublinear regret with respect to $T$, as this would indicate that the strategy $\pi^t$ effectively approaches the Nash equilibrium.

## 3 THEORY-MOTIVATED ALGORITHM

### 3.1 SETUP

We formulate the RLHF problem as a two-agent zero-sum game. Suppose that there exists a deterministic but unknown reward function $r^*(x, a)$ that represents the quality of response $a$ under prompt $x$. In practical applications, we want to ensure that the optimized policies $\pi$ and $\mu$ are close to a common reference policy $\pi_{\text{ref}}$. Therefore, we employ the following KL-regularized objective

$$\begin{aligned} V(\pi, \mu) = \mathbb{E}_{x \sim d_0, a^1 \sim \pi(\cdot|x), a^2 \sim \mu(\cdot|x)} \big[ &r^*(x, a^1) - r^*(x, a^2) \\ &- \alpha \cdot D_{\text{KL}}\left(\pi(\cdot|x)\|\pi_{\text{ref}}(\cdot|x)\right) + \alpha \cdot D_{\text{KL}}\left(\mu(\cdot|x)\|\pi_{\text{ref}}(\cdot|x)\right) \big]. \end{aligned} \tag{3}$$

Suppose we have access to a hypothesis class $\mathcal{R} \subset (\mathcal{X} \times \mathcal{A} \times \mathcal{A} \to \mathbb{R})$, which gives us a set of candidates to approximate the true reward function $r^*$. We define the value function under reward function $r$ as $V_r(\cdot, \cdot)$, and define $V_r(\pi, \dagger)$ as the value function induced by $\pi$ and its best response, i.e., $V_r(\pi, \dagger) = \min_\mu V_r(\pi, \mu)$.

## 3.2 Theoretical Algorithm Framework

Motivated by studies about exploration-exploitation trade-off from a practical perspective in traditional RL (Bellemare et al., 2016; Pathak et al., 2017), we propose our algorithm framework for online RLHF. The core idea is to optimize a single and unconstrained objective that simultaneously handles estimation and planning, thereby balancing exploration and exploitation. Specifically, the framework employs different strategies for each player: the max-player focuses on approximating the Nash equilibrium strategy, while the min-player aims to approximate the best response to the max-player's strategy. Both players plan with active exploration. In the $k$-th episode, the algorithm involves the following stages:

**At the first stage**, the max-player chooses the reward function $\hat{r}_t^1$ by maximizing the objective

$$\hat{r}_t^1 = \arg \max_{r \in \mathcal{R}} \left\{ V_r - \eta \cdot \mathcal{L}_{t-1}(r) \right\}. \tag{4}$$

To balance exploitation of historical data with exploration for the future, objective (4) consists of two components: **(i)** the negative loss function $-\mathcal{L}_{t-1}(r)$, the negative logistic regression loss in (2) computed on the data from the first $t-1$ episodes for the reward function $r$, to encourage exploitation, and **(ii)** the Nash equilibrium value $V_r$ associated with the reward function $r$, which promotes active exploration for the player. The algorithm balances the exploitation and exploration through a hyperparameter $\eta$.

With the max-player reward function $\hat{r}_t^1$, the max-player policy is set to be the Nash equilibrium max-player policy with respect to $\hat{r}_t^1$, i.e.,

$$\pi^t = \arg \max_{\pi \in \Pi} \min_{\mu \in \Pi} V_{\hat{r}_t^1}(\pi, \mu) = \arg \max_{\pi \in \Pi} \min_{\mu \in \Pi} \max_{r \in \mathcal{R}} \left\{ V_r - \eta \cdot \mathcal{L}_{t-1}(r) \right\}. \tag{5}$$

**At the second stage**, after obtaining the max-player policy $\pi^t$, the min-player chooses another reward function $\hat{r}_t^2$ aiming to find the best response of the max-player policy. Specifically, it chooses by minimizing the target

$$\hat{r}_t^2 = \arg \min_{r \in \mathcal{R}} \left\{ V_r(\pi^t, \dagger) + \eta \cdot \mathcal{L}_{t-1}(r) \right\}. \tag{6}$$

Objective (6) also has two components: **(i)** the loss function $\mathcal{L}_{t-1}(r)$ computed on historical data for the reward function $r$ to encourage exploitation, and **(ii)** the best response value $V_r(\pi^t, \dagger)$ to encourage active exploration.

With the min-player reward function $\hat{r}_t^2$, the min-player policy is set to be the best response of min-player policy under reward function $\hat{r}_t^1$, i.e.,

$$\mu^t = \arg \min_{\mu \in \Pi} V_{\hat{r}_t^2}(\pi^t, \mu) = \arg \min_{\mu \in \Pi} \min_{r \in \mathcal{R}} \left\{ V_r(\pi^t, \dagger) + \eta \cdot \mathcal{L}_{t-1}(r) \right\}. \tag{7}$$

**At the final stage**, the max-player and min-player sample a batch of new responses $\{(a_i^1, a_i^2)\}_{i=1}^N$ conditioned on prompts $\{x_i\}_{i=1}^N$ from the joint policy $(\pi^t, \mu^t)$ respectively. By querying human feedback or AI annotations $\{y_i\}_{i=1}^N$, we construct a new dataset $\mathcal{D}_t = \{x_i, a_i^1, a_i^2, y_i\}_{i=1}^N$ to update the loss function $\mathcal{L}_t(r)$.

## 4 Equivalent and Implementation-friendly Algorithms

### 4.1 Two-Agent Nash Policy Optimizaion

In this section, we introduce easy-to-implement objectives and algorithms that are equivalent to the algorithm framework in Section 3.2.

**Objective for the Max-Player.** We first examine the max-player objective in (5). If the reward function class $\mathcal{R}$ satisfies Assumption 4, the minimax theorem applies to optimization problem (5).

We refer readers to Appendix C for a detailed discussion. Hence, we interchange the *max* and *min* operations in (5), yielding an equivalent objective given by

$$\max_{r \in \mathcal{R}} \max_{\pi \in \Pi} \min_{\mu \in \Pi} \{V_r - \eta \cdot \mathcal{L}_{t-1}(r)\} = \max_{r \in \mathcal{R}} \left\{ \max_{\pi \in \Pi} \{F(\pi; r)\} + \min_{\mu \in \Pi} \{-F(\mu; r)\} - \eta \cdot \mathcal{L}_{t-1}(r) \right\},$$
(8)

where

$$F(\pi; r) := \mathbb{E}_{x \sim d_0, a \sim \pi(\cdot|x)} \left[ r(x, a) - \alpha \cdot D_{\mathrm{KL}} \left( \pi(\cdot|x) \| \pi_{\mathrm{ref}}(\cdot|x) \right) \right]$$

for any policy $\pi \in \Pi$ and $r \in \mathcal{R}$.

We first solve the inner optimization problem in (8), which enjoys the following closed-form solution as discussed in Rafailov et al. (2024):

$$\pi_r(a|x) = \arg\max_{\pi \in \Pi} F(\pi; r) = \frac{1}{Z_r(x)} \cdot \pi_{\mathrm{ref}}(a|x) \exp(r(x, a)/\alpha),$$
(9)

where we denote the partition function of $\pi_r$ as $Z_r(x) = \mathbb{E}_{a \sim \pi_{\mathrm{ref}}(\cdot|x)}[\exp(r(x, a)/\alpha)]$. Adopting the reparametrization technique as in Rafailov et al. (2024), we express the reward function $r$ by the optimal solution $\pi_r$, i.e.,

$$r(x, a) = \alpha \log \left( \frac{\pi_r(a|x)}{\pi_{\mathrm{ref}}(a|x)} \right) + \alpha \log Z_r(x).$$
(10)

We observe that, for a fixed reward function $r$, the two inner optimization problems in (8) have the same structure but opposite signs, and thus they cancel each other out, leaving only negative loss function term. Consequently, the max-player only needs to minimize the loss function on the historical data. Therefore, if we adopt the negative log-likelihood loss, the max-player's objective coincides with the objective of DPO algorithm, i.e.,

$$\min_{\pi \in \Pi} \left\{ \mathcal{L}_{\max}(\pi) := \eta \cdot \mathcal{L}_{t-1} \left( \alpha \log \left( \frac{\pi(a|x)}{\pi_{\mathrm{ref}}(a|x)} \right) \right) \right\}.$$
(11)

**Objective for the Min-Player.** The min-player optimization is a bilevel optimization problem and can be formulated as (we omit the terms independent of $\mu$ and $r$)

$$\min_{\mu \in \Pi} \min_{r \in \mathcal{R}} \left\{ V_r(\pi^t, \mu) + \eta \cdot \mathcal{L}_{t-1}(r) \right\} = \min_{r \in \mathcal{R}} \min_{\mu \in \Pi} \left\{ V_r(\pi^t, \mu) + \eta \cdot \mathcal{L}_{t-1}(r) \right\}$$

$$= \min_{r \in \mathcal{R}} \left\{ \mathbb{E}_{x \sim d_0, a \sim \pi^t(\cdot|x)} [r(x, a)] + \eta \cdot \mathcal{L}_{t-1}(r) + \min_{\mu \in \Pi} \{-F(\mu; r)\} \right\}.$$
(12)

We note that the inner optimization has the same structure as (9) and therefore enjoys the same closed-form solution. By substituting (9) into (12), using the reparameter technique in (10) and omitting irrelevant terms, we transform (12) into a single-level optimization to obtain the following min-player objective

$$\min_{\mu \in \Pi} \left\{ \mathcal{L}_{\min}(\mu) := \alpha \cdot \mathbb{E}_{x \sim d_0, a \sim \pi^t(\cdot|x)} [\log \mu(a|x)] + \eta \cdot \mathcal{L}_{t-1} \left( \alpha \log \left( \frac{\mu(a|x)}{\pi_{\mathrm{ref}}(a|x)} \right) \right) \right\}.$$
(13)

Hence, we summarize the Two-Agent Nash Policy Optimization (TANPO) algorithm in Algorithm 1.

---

**Algorithm 1** Two-Agent Nash Policy Optimization (TANPO)

---

**Input:** Reference policy $\pi_{\text{ref}}(\cdot)$, baseline policies $\pi^1(\cdot), \mu^1(\cdot)$ and parameters $\alpha, \eta$.
1: **for** $t = 1, 2, \ldots, T$ **do**
2:     Sample $a_i^1 \sim \pi^t(\cdot), a_i^2 \sim \mu^t(\cdot)$ from updated policies for each prompt $x_i$.
3:     Rank responses $a_i^1, a_i^2$ to form training dataset $\mathcal{D}_t = \{x_i, a_i^+, a_i^-\}_{i=1}^N$
4:     Update max-player policy according to (14),

$$\pi^{t+1} \leftarrow \arg\min_{\pi \in \Pi} \left\{ \eta \cdot \mathcal{L}\left( \alpha \log\left( \frac{\pi(\cdot|\cdot)}{\pi_{\text{ref}}(\cdot|\cdot)} \right) \middle| \mathcal{D}_t \right) \right\}. \tag{14}$$

5:     Update min-player policy according to (15),

$$\mu^{t+1} \leftarrow \arg\min_{\mu \in \Pi} \left\{ \eta \cdot \mathcal{L}\left( \alpha \log\left( \frac{\mu(\cdot|\cdot)}{\pi_{\text{ref}}(\cdot|\cdot)} \right) \middle| \mathcal{D}_t \right) + \alpha \cdot \mathbb{E}_{x \sim d_0, a \sim \pi^{t+1}(\cdot|x)} \left[ \log \mu(a|x) \right] \right\}, \tag{15}$$

    where $\mathcal{L}(\cdot|\cdot)$ denotes the logistic regression loss in (2).
6: **end for**

---

## 4.2 Data Diversity and Single-Agent Approximation

It is observed that although the min-player in TANPO benefits from an exploration bonus term, the max-player's objective in TANPO remains identical to DPO objective. This raises a question: How does TANPO offer improvements over DPO?

The key lies in the fact that TANPO generates more diverse training data. While the max-player does not have an explicit exploration bonus, it can still benefit from the increased diversity in the training data. In the original two-agent setup, both the max-player and min-player are trained on the same dataset but pursue different optimization objectives. The max-player focuses solely on minimizing the MLE loss function, aiming to maximize reward while closely approximate the reference policy $\pi_{\text{ref}}$. In contrast, the min-player incorporates an additional exploration bonus term $\mathbb{E}\left[ \log \mu(a|x) \right]$ into its objective, encouraging it to sample from less likely regions of the action distribution. As a result, the response pairs $(a^1, a^2)$ exhibit greater diversity and contrast.

Empirically, we demonstrate that TANPO leads to more diverse training data. To illustrate this, we sample 500 response pairs from the training datasets of online DPO and TANPO, respectively. For each response pair $(a^1, a^2)$, we calculate the difference in their length-normalized log probabilities under the reference policy, $|\log \pi_{\text{ref}}(a^1) - \log \pi_{\text{ref}}(a^2)|$, as a measure of diversity between responses. As shown in Figure 1, TANPO achieves a larger $\log \pi_{\text{ref}}$ margin between response pairs, indicating increased diversity in the training data. These findings further confirm that TANPO enhances data diversity, thereby improving the overall performance.

To simplify this setup, we propose the Single-Agent Diversity-driven Policy Optimization (SADPO) algorithm as a single-agent approximation of TANPO. The SADPO optimization objective is similar to min-player objective (13). The core idea is to use a single policy to simulate the max-player policy and min-player policy through rejection sampling. At each training iteration, the agent samples $K$ responses from the current policy for each prompt. We then compute the probabilities $\pi_{\text{ref}}(a)$ for each of the $K$ samples. From the $K$ responses, the response with the highest $\pi_{\text{ref}}$ value is selected to approximate the behavior of the max-player, as this response is the most aligned with the reference policy. The response with the lowest $\pi_{\text{ref}}$ value is chosen to represent the min-player's behavior, as it reflects more exploratory responses with lower likelihood under $\pi_{\text{ref}}$. To mitigate the effect of response length, we calculate $\log \pi_{\text{ref}}$ by averaging the log probability over the response length. We summarize SADPO in Algorithm 2.

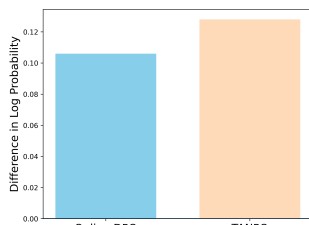

Figure 1: Length-normalized ref. policy log probability difference between response pairs.

An important feature of SADPO is that it deliberately enlarges the difference in $\pi_{\text{ref}}$ between the selected responses. By systematically selecting the most and least

likely responses according to $\pi_{\text{ref}}$, SADPO increases the diversity of the training data. This helps the model explore a broader range of behaviors, ultimately improving its generalization and overall performance.

---

**Algorithm 2** Single-Agent Diversity-driven Policy Optimization (SADPO)

---

**Input:** Reference policy $\pi_{\text{ref}}(\cdot)$, baseline policy $\pi^1(\cdot)$ and parameters $\alpha, \eta, K$.

1: **for** $t = 1, 2, \ldots, T$ **do**
2:      Sample $K$ responses $\{a_i^k\}_{k=1}^K \sim \pi^t(\cdot)$ for each prompt $x_i$.
3:      For each $a_i^k$, compute its probability under the reference policy $\pi_{\text{ref}}(a_i^k|x_i)$.
4:      Select two responses by
       • $a_i^{\max} = \arg\max_k \pi_{\text{ref}}(a_i^k|x_i),$           ▷ approximating max-player behavior
       • $a_i^{\min} = \arg\min_k \pi_{\text{ref}}(a_i^k|x_i).$           ▷ approximating min-player behavior
5:      Rank responses $a_i^{\max}, a_i^{\min}$ to form training dataset $\mathcal{D}_t = \{x_i, a_i^+, a_i^-\}_{i=1}^N$.
6:      Update the policy according to (16),

$$\pi^{t+1} \leftarrow \arg\min_{\pi \in \Pi} \left\{ \eta \cdot \mathcal{L}\left( \alpha \log\left( \frac{\pi(\cdot|\cdot)}{\pi_{\text{ref}}(\cdot|\cdot)} \right) \Big| \mathcal{D}_t \right) + \alpha \cdot \mathbb{E}_{x \sim d_0, a \sim \pi_{\text{ref}}(\cdot|x)}\left[ \log \pi(a|x) \right] \right\}, \tag{16}$$

     where $\mathcal{L}(\cdot|\cdot)$ denotes the logistic regression loss as defined in (2).
7: **end for**

---

## 5 THEORETICAL ANALYSIS

In this section, we present the regret analysis for the theoretical two-player Nash RLHF algorithm introduced in Section 3.2. We first provide theoretical guarantees under the low TGEC conditions (Assumption 2). Next, we illustrate our results using a linear two-player zero-sum RLHF game as a concrete example. It is important to note that the theoretical analysis in this section also applies to TANPO (Algorithm 1), provided the reward function class $\mathcal{R}$ meets Assumption 4 outlined in Appendix C.

### 5.1 REGRET ANALYSIS FOR TWO-PLAYER NASH RLHF

To derive the theorem, we first present two assumptions. The first assumption concerns the hypothesis class $\mathcal{R}$ being finite, bounded and well-specified, meaning it contains the true hypothesis.

**Assumption 1** (Realizability). *We assume that the true reward model $r^* \in \mathcal{R}$, and $\mathcal{R}$ is finite, i.e. $|\mathcal{R}| < +\infty$. Moreover, for regularization, we assume that the reward function is bounded: for any $r \in \mathcal{R}$ and any $(x, a^1, a^2) \in \mathcal{X} \times \mathcal{A} \times \mathcal{A}$, we have $|r(x, a^1) - r(x, a^2)| \le R_0$ for some $R_0 > 0$.*

Moreover, we make a structural assumption on the underlying two-player game that requires the game to have a low **Two-player Generalized Eluder Coefficient** (TGEC) $d_{\text{TGEC}}(\cdot)$. TGEC is the generalization of Generalized Eluder Coefficient (GEC) inspired by Zhong et al. (2022). In a two-agent Markovian game with low TGEC, the two agents can minimize the in-sample prediction error on historical data, thereby decreasing the out-of-sample prediction error. See the full statement in Assumption 2 in Appendix B.

With Assumptions 1 and 2, we now present our main theorem.

**Theorem 1** (Online Regret of Two-agent Nash RLHF). *Under Assumptions 1 and 2, by setting*

$$\eta = \frac{1}{4}\sqrt{\frac{d_{\text{TGEC}}(1/\sqrt{T})}{T \log(|\mathcal{R}|/\delta)}},$$

*the regret of our theoretical algorithm framework in Section 3.2 after $T$ episodes is bounded by*

$$\text{Regret}(T) \le 2\left( \sqrt{d_{\text{TGEC}}(1/\sqrt{T})\log(|\mathcal{R}|/\delta)} + \sqrt{d_{\text{TGEC}}(1/\sqrt{T})} + 1 \right)\sqrt{T}$$

*with probability at least $1 - 2\delta$.*

Theorem 1 provides a theoretical guarantee for the efficiency of the algorithm's learning process in RLHF problems that satisfy the low TGEC condition. When the iteration number $T$ tends to infinity, the average regret $\mathrm{Regret}(T)/T$ tends to zero. This indicates that the resulting policy of TANPO is approximately a Nash equilibrium policy, demonstrating the sample efficiency of TANPO.

As a concrete example of Theorem 1, we further analyze the case of a linear two-player Nash RLHF game. In this setting, we explicitly compute the TGEC bound and establish a sublinear regret guarantee for the max-player in Corollary 1. This demonstrates that our theoretical framework not only holds in general settings but also provides clear and provable efficiency in specific and structured cases like linear games. We refer readers to Appendix B for detailed discussion.

## 6 EXPERIMENTS

In this section, we conduct detailed experiments to show the performances of our practical algorithms along with other baselines. Our experiments demonstrate three key findings: **(i)** Our algorithms consistently outperform baseline methods across various benchmarks. **(ii)** Our algorithms effectively mitigate overfitting. **(iii)** Our algorithms enhance model performance by improving the quality and diversity of the training data.

### 6.1 EXPERIMENT SETUP

We use UltraFeedback (Cui et al., 2023) as our training dataset, which consists of 61k high-quality prompts and response pairs annotated by GPT-4. We split the UltraFeedback dataset into three portions, and use only one portion on each iteration. We first conduct an offline DPO training on the first portion of training dataset, and then conduct two iterations of online alignment on the other two portions. For the base model of our training, we consider Zephyr series of LLMs (Tunstall et al., 2023). We choose Zephyr-7B-SFT as our base model, since the official Zephyr-7B-$\beta$ has already been fine-tuned on the same UltraFeedback dataset. The small-sized PairRM (Jiang et al., 2023) is used as the preference model to provide AI feedback during online alignment.

In TANPO, we sample one response for each prompt from each of the two models. In SADPO, we sample $K = 4$ responses for each prompt from the current policy, and select the responses with the highest and lowest length-regularized reference policy log probability to form response pairs. We choose the base model Zephyr-7B-SFT, online DPO, Hybrid GSHF (Xiong et al., 2023) and SELM (Zhang et al., 2024) as baselines for fine-tuning LLM. For evaluation, we adopt AlpacaEval 2.0 (Dubois et al., 2024), MT-Bench (Zheng et al., 2023) and several academic benchmarks, including GSM8k (Cobbe et al., 2021), MMLU (Hendrycks et al., 2020), OpenBookQA (Mihaylov et al., 2018), HellaSwag (Zellers et al., 2019) and WinoGrande (Sakaguchi et al., 2021). All the implementation details are provided in Appendix E.

### 6.2 EXPERIMENT RESULTS

**Our algorithms consistently improve the performance on different benchmarks.** Our algorithms consistently improve the performance on different benchmarks. We present our main result in Table 1. Here, we report the results of TANPO based on the performance of the min-player. The full results, including the performance of both players in TANPO, are provided in Table 2 in Appendix D. On AlpacaEval 2.0 results, TANPO achieves high performance, with a length-controlled win rate of 27.66% and a win rate of 27.08%, outperforming all baseline methods. Similarly, SADPO demonstrates strong results across all baselines, achieving a length-controlled win rate of 28.43% and a win rate of 26.21%. Besides, we compare TANPO and SADPO with other online RLHF baselines on MT-Bench scores. Notably, TANPO and SADPO achieve the first and second places respectively in terms of average MT-Bench scores. Results on several academic benchmarks presented in Figure 2 show that our methods outperform the baselines on average and across the majority of academic benchmarks. The full results are reported in Table 3 in Appendix

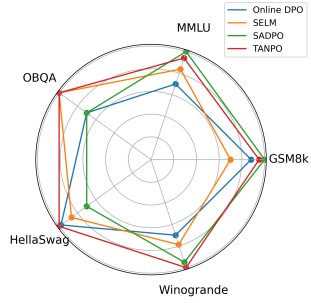

Figure 2: Accuracy results on GSM8k, MMLU, OpenBookQA, HellaSwag and WinoGrande.

D. Furthermore, we examine the pairwise win rate among baseline models, single-agent models and two-agent models, using PairRM as the judge on 805 prompts from the AlpacaEval 2.0 dataset. The results are depicted in Figure 3.

| Technique | AlpacaEval 2.0 | | MT-Bench | | |
|---|---|---|---|---|---|
| | LC Win Rate | Win Rate | Average | 1st Turn | 2nd Turn |
| Zephyr-7B-SFT (ref.) | 6.59 | 3.66 | 6.14 | 6.34 | 5.95 |
| Online DPO | 24.36 | 22.14 | 7.24 | 7.37 | 7.11 |
| Hybrid GSHF | 25.29 | 22.61 | 7.28 | 7.26 | 7.30 |
| SELM | 26.99 | 25.99 | 7.26 | 7.56 | 6.96 |
| SADPO | **28.43** | 26.21 | 7.33 | **7.71** | 6.94 |
| TANPO | 27.66 | **27.08** | **7.47** | 7.55 | **7.39** |

Table 1: Results on AlpacaEval 2.0 and MT-Bench. LC Win Rate represents Length-Controlled Win Rate.

**Our algorithms effectively mitigate overfitting.** To further evaluate the performance of our proposed algorithm and investigate its robustness against overfitting, we conduct a second round of experiments on the same UltraFeedback dataset. Specifically, after completing the first three iterations, we continue training the model with additional three iterations on the same dataset, while monitoring AlpacaEval 2.0 metrics. Our results are shown in Figure 4, demonstrating that the model continues to improve during the second round of training, showing no signs of overfitting. The full results are presented in Table 4 in Appendix D. This suggests that our two-agent algorithm is capable of effectively utilizing the training data, even after an extended training period. We note that our algorithm optimizes two agents using distinct strategies while having them compete against each other throughout the training process. This competition leads to a natural increase in the diversity of the training data, as each agent generates responses based on different optimization paths, resulting in more varied and comprehensive scenarios. Additionally, the algorithm's active exploration mechanism prevents the agents from getting stuck in local minima. These findings highlight the ability of our approach in consistently improving performance and preventing overfitting.

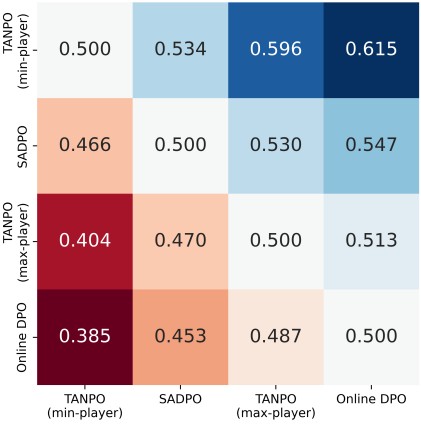

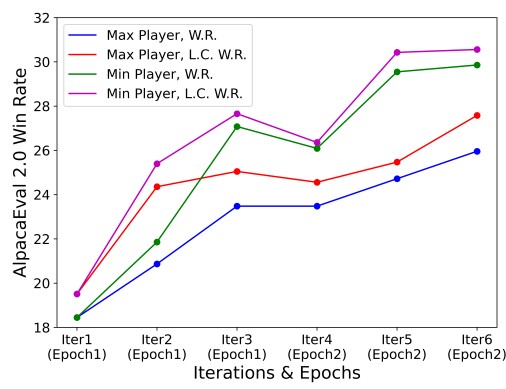

Figure 3: Pairwise win rates among baseline models, single-agent models and two-agent models, using PairRM as a judge.

Figure 4: Performance of TANPO across 6 iterations (2 epochs) on win rate (W.R.) and length-controlled win rate (L.C. W.R.) judged by GPT-4-Turbo.

**Our algorithms enhance model performance by improving the quality and diversity of the training data.** Our results show that TANPO (max-player) surpasses the online DPO by 0.69% and 1.34% in AlpacaEval 2.0 length-controlled win rate and win rate. Besides, TANPO (max-player) achieves a win rate of 51.3% against online DPO in pairwise comparison by PairRM. It is important to note that the max-player shares the same setup as the online DPO, with the only difference being the training data. We have shown in Figure 1 that TANPO achieves greater diversity in training

data than online DPO. These results suggest that our algorithm enhances model performance by increasing the diversity of responses during training. By incorporating data generated from two different policies, the model is exposed to a broader range of behaviors and strategies, which helps it generalize better across different prompts.

# 7 CONCLUSION

In this work, we introduce Two-Agent Nash Policy Optimization (TANPO), a two-agent algorithm that balances exploration and exploitation. Additionally, we present Single-Agent Diversity-driven Optimization (SADPO) as a simplified approximation of TANPO, supported by theoretical and empirical results. Our theoretical analysis shows sublinear regret under general conditions, while empirical evaluations demonstrate that TANPO and SADPO outperform baseline methods across multiple benchmarks, highlighting their effectiveness in improving performance and reducing overfitting. We hope our work can provide insights for future research into designing provable efficient and practical self-play RLHF methods.

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

# A  RELATED WORKS

## A.1  THEORETICAL STUDIES ABOUT RLHF

Theoretical research on RLHF largely originates from foundational work in dueling bandits and dueling reinforcement learning (Xu et al., 2020; Novoseller et al., 2020; Pacchiano et al., 2021; Wu & Sun, 2023; Zhan et al., 2023). Recently, there has been growing interest in exploring RLHF theory across diverse settings. Many previous works only consider tabular settings and linear settings (Du et al., 2024; Das et al., 2024; Xiong et al., 2023), which do not fully capture the complexities of real-world scenarios. Beyond the tabular and linear settings, some recent research has explored RLHF theory under general function approximation (Chen et al., 2022; Wang et al., 2023; Zhu et al., 2023; Liu et al., 2024b).

Among these researches, most approaches use a contextual bandit framework. Close to our approach, some studies give algorithms and theoretical guarantees under two-agent zero-sum game setting (Wang et al., 2023; Ye et al., 2024). Ye et al. (2024) considered a reverse-KL regularized minimax game between two LLMs and gave a sample-efficient algorithm, and Wang et al. (2023) gave guarantees for RLHF algorithm under general arbitrary preferences settings. However, their algorithms contain theoretical confidence bounds that are hard to compute, thus are unable to be implemented in experiments. Liu et al. (2024a) proposed a two-agent algorithm framework which approximates Nash equilibrium with active exploration and proved the algorithm has a sublinear guarantee. Different from ours, their work only focused on MDP and Markov game settings.

## A.2  EMPIRICAL ALGORITHMS FOR RLHF

RLHF has gained significant traction in the deployment of large-scale models, such as ChatGPT (Achiam et al., 2023), Gemini (Team et al., 2023) and Llama (Touvron et al., 2023), where it has been employed to align model behavior with human preferences effectively. A widely used algorithm in this context is Proximal Policy Optimization (PPO) (Schulman et al., 2017), which iteratively updates the model feedback from a learned reward model. However, while PPO has demonstrated success in many practical applications, it also has notable limitations, such as high sample complexity and sensitivity to hyperparameters (Engstrom et al., 2020), which can affect its stability and performance. To address these shortcomings, recent work has proposed alternative approaches like Direct Preference Optimization (DPO) (Rafailov et al., 2024), which directly optimizes the model based on human preferences. In addition to the initial formulation of the DPO algorithm, researchers have developed a wide range of variants (Liu et al., 2023; Azar et al., 2024; Ethayarajh et al., 2024; Hong et al., 2024; Liu et al., 2024b), each tailored to specific optimization problems or designed to enhance certain aspects of the algorithm's performance.

Unlike offline RLHF, which is constrained by a static dataset, online RLHF (Ouyang et al., 2022; Guo et al., 2024; Bai et al., 2022; Xu et al., 2023; Gulcehre et al., 2023; Xiong et al., 2023; Calandriello et al., 2024; Pang et al., 2024; Sun et al., 2024; Chen et al., 2024; Ji et al., 2024) continually generates better and more diverse learning data as the model learns and adapts over time, allowing the model to refine its understanding through real-time human feedback. In online RLHF, exploration plays a critical role by allowing the model to avoid being trapped in local minima, continuously encouraging it to seek new actions and learn from a more diverse set of responses. Zhang et al. (2024); Xie et al. (2024); Cen et al. (2024) propose algorithms that add DPO with exploration bonuses similar to ours. Liu et al. (2024b) also utilize a similar confidence bound but aim to avoid overoptimization in offline RLHF. Unlike our work, these works only focus on single-agent contextual bandit settings.

## B  THEORETICAL ANALYSIS

To conduct the theoretical analysis for our theoretical algorithm framework proposed in Section 3.2, we introduce a more general reward model under which the reward model (1) is a special case.

**Preference Model.**  Given any reward function $R : \mathcal{X} \times \mathcal{A} \times \mathcal{A} \to \mathbb{R}$ which represents the "human's rating" of LLM responses given some prompts, we denote that given a prompt $x \in \mathcal{X}$ and two response $a^1, a^2 \in \mathcal{A}$, the probability of $a^1$ being preferred to $a^2$ (denoted by $y = 1$, and otherwise by $y = 0$) is given by

$$\mathbb{P}_R(y = 1 | x, a^1, a^2) = \sigma(R(x, a^1, a^2)), \tag{17}$$

where $\sigma(z) = 1/(1 + \exp(-z))$ is the sigmoid function. Clearly, $\mathbb{P}_R(y = 0 | x, a^1, a^2) = \sigma(-R(x, a^1, a^2))$. We denote the underlying reward model as $R^*$, and the corresponding preference oracle as $\mathcal{P}^* := \mathbb{P}_{R^*}$. Notably, a special case for the reward function $R$ is by assuming that there exists a function $r : \mathcal{X} \times \mathcal{A} \to \mathbb{R}$ such that

$$R(x, a^1, a^2) = r(x, a^1) - r(x, a^2) \quad \text{for any } x \in \mathcal{X} \text{ and } a^1, a^2 \in \mathcal{A} \tag{18}$$

holds, which coincides with the reward model (1) in the main text.

**Nash Equilibrium and Best Response.**  As discussed in the main text, RLHF is formulated as a two-player Game between the max-player LLM $\pi$ and the min-player LLM $\mu$. In application, we want the resulting LLMs $(\pi, \mu)$ to be close to $(\pi_{\text{ref}}, \pi_{\text{ref}})$, and the ultimate value function is given by

$$V_{R^*, \alpha, \beta}(\pi, \mu) = \mathbb{E}_{x \sim d_0, a^1 \sim \pi(\cdot|x), a^2 \sim \mu(\cdot|x)} \left[ R^*(x, a^1, a^2) \right.$$
$$\left. - \alpha \cdot D_{\text{KL}}(\pi(\cdot|x) \| \pi_{\text{ref}}(\cdot|x)) + \alpha \cdot D_{\text{KL}}(\mu(\cdot|x) \| \pi_{\text{ref}}(\cdot|x)) \right]. \tag{19}$$

For simplicity, we denote $V_{R^*, \alpha, \beta}(\cdot, \cdot)$ as $V(\cdot, \cdot)$ when there is no ambiguity. The value function (19) under the reward model $R$ coincides with the value function (3) under the reward model $r$ by (18). We denote the unique Nash equilibrium (NE) as the solution of the following minimax problem as

$$(\pi^*, \mu^*) = \arg \max_{\pi \in \Pi} \arg \min_{\mu \in \Pi} V(\pi, \mu).$$

For function $V$ and policy $\pi$, the best response to $\pi$ is $\arg \min_{\mu \in \Pi} V(\pi, \mu)$, and the value is denoted by $V(\pi, \dagger) = \min_{\mu \in \Pi} V(\pi, \mu)$. Similarly, for $\mu$, we have $V(\dagger, \mu) = \max_{\pi \in \Pi} V(\pi, \mu)$.

**Function Approximation.**  For convenience, we introduce the following notations. We have access to a function class $\mathcal{R} \subset (\mathcal{X} \times \mathcal{A} \times \mathcal{A} \to \mathbb{R})$ to approximate $R^*$. Specifically, given parameter $\alpha$, for any $R \in \mathcal{R}$:

1. we denote the corresponding reward function as $R(x, a^1, a^2)$, and corresponding value function as

$$V_R(\pi, \mu) = \mathbb{E}_{x \sim d_0, a^1 \sim \pi(\cdot|x), a^2 \sim \mu(\cdot|x)} \left[ R(x, a^1, a^2) \right.$$
$$\left. - \alpha \cdot D_{\text{KL}}(\pi(\cdot|x) \| \pi_{\text{ref}}(\cdot|x)) + \alpha \cdot D_{\text{KL}}(\mu(\cdot|x) \| \pi_{\text{ref}}(\cdot|x)) \right];$$

2. we denote the corresponding NE value function as

$$V_R = \max_{\pi \in \Pi} \min_{\mu \in \Pi} V_R(\pi, \mu), \tag{20}$$

and we denote the corresponding NE max-player policy as

$$\pi_R = \arg \max_{\pi \in \Pi} \min_{\mu \in \Pi} V_R(\pi, \mu); \tag{21}$$

3. given a policy as the max-player $\pi \in \Pi$, we define $V_R(\pi, \dagger)$ as the value function induced by $R, \pi$ and its best response, i.e.

$$V_R(\pi, \dagger) = \min_{\mu \in \Pi} V_R(\pi, \mu), \tag{22}$$

and we denote the corresponding best response min-player policy as

$$\mu_{R, \pi} = \arg \min_{\mu \in \Pi} V_R(\pi, \mu); \tag{23}$$

---

**Algorithm 3** Theoretical Algorithm

---

**Input:** Parameters $\alpha, \beta, \eta > 0$, prompt distribution $d_0$, time horizon $T$, reference policy $\pi_{\text{ref}}$.
1: Set $\pi^0 = \mu^0 = \pi_{\text{ref}}$.
2: **for** step $t = 0, 1, 2, \cdots, T$ **do**
3:     Define the loss function $\mathcal{L}_{t-1}(R)$ in (28).
4:     Solve $R_1^t$ via

$$R_1^t = \arg\max_{R \in \mathcal{R}} \left\{ V_R - \eta \cdot \mathcal{L}_{t-1}(R) \right\}. \tag{24}$$

5:     Set the max-player policy as

$$\pi^t = \pi_{R_1^t}. \tag{25}$$

6:     Solve $R_2^t$ via

$$R_2^t = \arg\min_{R \in \mathcal{R}} \left\{ V_R(\pi^t, \dagger) + \eta \cdot \mathcal{L}_{t-1}(R) \right\}. \tag{26}$$

7:     Set the min-player policy as

$$\mu^t = \mu_{R_2^t, \pi^t}. \tag{27}$$

8:     Collect $\mathcal{D}_t = \{x_t, a_t^1, a_t^2, y_t\}$ by $x_t \sim d_0, a_t^1 \sim \pi^t(\cdot|x_t), a_t^2 \sim \mu^t(\cdot|x_t), y_t \sim \mathcal{P}^*(\cdot|x_t, a_t^1, a_t^2)$.
9: **end for**

---

    4. we denote the NE value function under the true reward $R^*$ as $V_{R^*}$.

With the reward model (18), the theoretical algorithm framework in Section 3.2 can be rewritten into Algorithm 3, where we specify the loss function in the $t$-th episode as the negative log-likelihood function of the preference model (17), defined as

$$
\begin{aligned}
\mathcal{L}_{t-1}(R) &= -\sum_{s=1}^{t-1} \log \mathbb{P}_R \left( y = y_s | x_s, a_s^1, a_s^2 \right) \\
&= -\sum_{s=1}^{t-1} \left[ y_s \cdot \log \left( \sigma \left( R(x_s, a_s^1, a_s^2) \right) \right) + (1 - y_s) \cdot \log \left( \sigma \left( -R(x_s, a_s^1, a_s^2) \right) \right) \right],
\end{aligned}
\tag{28}
$$

which coincides with the logistic regression loss (2) by (18). Therefore, to prove Theorem 1, it suffices to analyze the regret bound of Algorithm 3.

To define TGEC, we first introduce the discrepancy function $l(R; \xi) : \mathcal{R} \times (\mathcal{X} \times \mathcal{A} \times \mathcal{A}) \mapsto \mathbb{R}$ to characterize the bellman residuals of both players. We choose Hellinger distance as the discrepancy function. For any data $\xi = (x, a^1, a^2)$, we define

$$l(R; \xi) = D_{\text{H}} \left( \mathcal{P}^*(\cdot|\xi) \| \mathbb{P}_R(\cdot|\xi) \right), \tag{29}$$

where $D_{\text{H}}(\cdot\|\cdot)$ denotes the Hellinger distance: for two discrete probability distributions $P = (p_1, \ldots, p_k)$ and $Q = (q_1, \ldots, q_k)$, the Hellinger distance is defined as

$$D_{\text{H}}(P\|Q) = \frac{1}{2} \sum_{i=1}^{k} (\sqrt{p_i} - \sqrt{q_i})^2. \tag{30}$$

**Assumption 2** (Low Two-Player Generalized Eluder Coefficient)**.** *Given any $\epsilon > 0$, there exists a finite $d(\epsilon) \in \mathbb{R}_+$, such that for any sequence $\{(R_1^t, R_2^t)\}_{t=1}^T$ and corresponding policies $\{\pi^t, \mu^t\}_{t=1}^T$ by (25) and (27) respectively, it holds that*

$$\sum_{t=1}^{T} \left[ V_{R_1^t} - V(\pi^t, \mu^t) \right] \leq \inf_{\zeta > 0} \left\{ \frac{\zeta}{2} \sum_{t=1}^{T} \sum_{s=1}^{t-1} \mathbb{E}_{\xi_s \sim (d_0, \pi^s, \mu^s)} \left[ l(R_1^t; \xi_s) \right] + \frac{d(\epsilon)}{2\zeta} + \sqrt{d(\epsilon)T} + \epsilon T \right\}.$$

*It also holds that*

$$\sum_{t=1}^{T} \left[ V(\pi^t, \mu^t) - V_{R_2^t}(\pi^t, \dagger) \right] \leq \inf_{\zeta > 0} \left\{ \frac{\zeta}{2} \sum_{t=1}^{T} \sum_{s=1}^{t-1} \mathbb{E}_{\xi_s \sim (d_0, \pi^s, \mu^s)} \left[ l(R_2^t; \xi_s) \right] + \frac{d(\epsilon)}{2\zeta} + \sqrt{d(\epsilon)T} + \epsilon T \right\}.$$

*We denote the smallest $d(\epsilon) \in \mathbb{R}_+$ satisfying this condition as $d_{\text{TGEC}}(\epsilon)$.*

Moreover, by taking the decomposable reward function in (3), through some basic algebra we can derive the equivalent TGEC assumption under Nash RLHF setting.

**Assumption 3** (Low Two-player Generalized Eluder Coefficient under Nash RLHF). *Given any $\epsilon > 0$, there exists a finite $\widetilde{d}(\epsilon) \in \mathcal{R}_+$, such that for any sequence $\{(R_1^t, R_2^t)\}_{t=1}^T$ with form of*

$$R_1^t(x, a, b) = r_1^t(x, a) - r_1^t(x, b), \ \ R_2^t(x, a, b) = r_2^t(x, a) - r_2^t(x, b)$$

*and corresponding policies $\{\pi^t, \mu^t\}_{t=1}^T$ by (25) and (27) respectively, it holds that*

$$-\sum_{t=1}^T \mathbb{E}_{(x,a_1,a_2)\sim(d_0,\pi^t,\mu^t)} \left[ r^*(x, a^1) - r^*(x, a^2) + \alpha \left[ \log\left( \frac{\mu^t(a^2|x)}{\pi_{\text{ref}}(a^2|x)} \right) - \log\left( \frac{\pi^t(a^1|x)}{\pi_{\text{ref}}(a^1|x)} \right) \right] \right]$$

$$\leq \inf_{\zeta>0} \left\{ \frac{\zeta}{2} \sum_{t=1}^T \sum_{s=1}^{t-1} \mathbb{E}_{(x,a_1,a_2)\sim(d_0,\pi^t,\mu^t)} \left[ l(R_1^t; \xi_s) \right] + \frac{d(\epsilon)}{2\zeta} + \sqrt{d(\epsilon)T} + \epsilon T \right\}.$$

*It also holds that*

$$\sum_{t=1}^T \mathbb{E}_{(x,a_1,a_2)\sim(d_0,\pi^t,\mu^t)} \left[ r^*(x, a^1) - r^*(x, a^2) + \alpha \left[ \log\left( \frac{\mu^t(a^2|x)}{\pi_{\text{ref}}(a^2|x)} \right) - \log\left( \frac{\mu^t(a^1|x)}{\pi_{\text{ref}}(a^1|x)} \right) \right] \right]$$

$$\leq \inf_{\zeta>0} \left\{ \frac{\zeta}{2} \sum_{t=1}^T \sum_{s=1}^{t-1} \mathbb{E}_{(x,a_1,a_2)\sim(d_0,\pi^t,\mu^t)} \left[ l(R_1^t; \xi_s) \right] + \frac{d(\epsilon)}{2\zeta} + \sqrt{d(\epsilon)T} + \epsilon T \right\}.$$

### B.1 Linear Two-Player Zero-Sum Nash RLHF Game

In this section, we introduce the linear two-player Nash RLHF game as a concrete example (Xie et al., 2020), for which we can explicitly specify its TGEC, thus specify Theorem 1 for linear two-player Nash RLHF game case.

**Definition 1** (Linear two-player zero-sum Nash RLHF game). *A $d$-dimensional linear two-player zero-sum Nash RLHF game satisfies that $R(x, a^1, a^2) = \phi(x, a^1, a^2)^\top \lambda$ for some known feature mapping $\phi(x, a^1, a^2) \in \mathbb{R}^d$ and some unknown vector $\lambda \in \mathbb{R}^d$ satisfying $\|\phi(x, a^1, a^2)\|_2 \leq 1$ and $\|\lambda\|_2 \leq \sqrt{d}$ for any $(x, a^1, a^2) \in \mathcal{X} \times \mathcal{A} \times \mathcal{A}$.*

For a linear two-player zero-sum Nash RLHF game, we choose the reward hypothesis class as

$$\mathcal{R} = \left\{ \phi(\cdot, \cdot, \cdot)^\top \lambda : \|\lambda\|_2 \leq \sqrt{d} \right\}. \tag{31}$$

The following proposition gives the TGEC of a linear two-player zero-sum Nash RLHF game with hypothesis class $\mathcal{R}$.

**Proposition 1** (TGEC of linear two-player zero-sum Nash RLHF game). *For a linear two-player zero-sum Nash RLHF game with hypothesis class $\mathcal{R}$, it holds that*

$$d_{\text{TGEC}}(1/\sqrt{T}) \leq 4\kappa^2 d \cdot \log\left( 1 + \frac{T^{2/3}}{d} \right) \lesssim d\log T,$$

*where the constant $\kappa = (e^{R_0/2} + e^{-R_0/2})^2$.*

Thus, we can specify Theorem 1 for linear two-player zero-sum games as follows.

**Corollary 1** (Online regret: linear two-player zero-sum Nash RLHF game). *By setting $\eta = \widetilde{\Theta}(1/\sqrt{T})$, the regret of our theoretical algorithm for a $d$-dimensional linear two-player zero-sum Nash RLHF game after $T$ episodes is upper bounded by*

$$\text{Regret}(T) \leq 2\sqrt{T} \left( 4\kappa^2 d \cdot \log\left( 1 + T^{2/3}/d \right) \log(|\mathcal{R}|/\delta) + 2\sqrt{2\max\{R_0, 1\}d \cdot \log\left( 1 + T^{2/3}/d \right)} + d \right)$$

*with probability at least $1 - \delta$ and the constant $\kappa = (e^{R_0/2} + e^{-R_0/2})^2$.*

### B.2 PROOF OF THEOREM 1

*Proof of Theorem 1.* This is a similar proof to that of Theorem 4.4 in Liu et al. (2024a). It suffices to analyze the regret bound of Theorem 3 under the general reward model (18). We first show a proposition for loss difference.

**Proposition 2.** *Under Assumption 1, with probability at least $1 - \delta$, for any $R \in \mathcal{R}$, it holds that*

$$\sum_{t=1}^{T} [\mathcal{L}_{t-1}(R^*) - \mathcal{L}_{t-1}(R)] \leq -2 \sum_{t=1}^{T} \sum_{s=1}^{t-1} \mathbb{E}_{\xi_s \sim (d_0, \pi^s, \mu^s)} [\ell(R; \xi_s)] + 2T \log(|\mathcal{R}|/\delta),$$

*where $\mathcal{L}_{t-1}$ and $\ell$ are defined in (28) and (29) respectively.*

*Proof of proposition 2.* This is a similar proof to that of Proposition C.13 in Liu et al. (2024a). Given $R \in \mathcal{R}$, we denote the random variables $X_R^s$ as

$$X_R^s = \log \frac{\mathcal{P}^*(y = y_s | x_s, a_s^1, a_s^2)}{\mathbb{P}_R(y = y_s | x_s, a_s^1, a_s^2)}. \tag{32}$$

By the definition of $\mathcal{L}_{t-1}$ in (28), we have

$$\sum_{t=1}^{T} [\mathcal{L}_{t-1}(R^*) - \mathcal{L}_{t-1}(R)] = -\sum_{t=1}^{T} \sum_{s=1}^{t-1} X_R^s.$$

We next define a filtration $\{\mathcal{F}_t\}_{t=1}^T$ for each $t \in [T]$ with

$$\mathcal{F}_t = \sigma \left( \bigcup_{s=1}^{t} \mathcal{D}_s \right).$$

Then by (32) we have that $X_R^t \in \mathcal{F}_t$ for each $t \in [T]$. Therefore, by Lemma B.3, with probability at least $1 - \delta$, for any $R \in \mathcal{R}$ and $t \in [T]$, we have that

$$-\frac{1}{2} \sum_{s=1}^{t-1} X_R^s \leq \sum_{s=1}^{t-1} \log \mathbb{E} \left[ \exp \left( -\frac{1}{2} X_R^s \right) \Big| \mathcal{F}_{s-1} \right] + \log(|\mathcal{R}|/\delta). \tag{33}$$

Moreover, the conditional expectation in (33) has the expression

$$\mathbb{E} \left[ \exp \left( -\frac{1}{2} X_R^s \right) \Big| \mathcal{F}_{s-1} \right]$$

$$= \mathbb{E} \left[ \sqrt{\frac{\mathbb{P}_R(y = y_s | x_s, a_s^1, a_s^2)}{\mathcal{P}^*(y = y_s | x_s, a_s^1, a_s^2)}} \Big| \mathcal{F}_{s-1} \right]$$

$$= \mathbb{E}_{\substack{x_s \sim d_0, a_s^1 \sim \pi^s(\cdot|x_s), a_s^2 \sim \mu^s(\cdot|x_s) \\ y_s \sim \mathcal{P}^*(\cdot|x_s, a_s^1, a_s^2)}} \left[ \sqrt{\frac{\mathbb{P}_R(y = y_s | x_s, a_s^1, a_s^2)}{\mathcal{P}^*(y = y_s | x_s, a_s^1, a_s^2)}} \right]$$

$$= \mathbb{E}_{x_s \sim d_0, a_s^1 \sim \pi^s(\cdot|x_s), a_s^2 \sim \mu^s(\cdot|x_s)} \left[ \sum_{y_s = 0, 1} \sqrt{\mathbb{P}_R(y = y_s | x_s, a_s^1, a_s^2) \mathcal{P}^*(y = y_s | x_s, a_s^1, a_s^2)} \right]$$

$$= 1 - \frac{1}{2} \mathbb{E}_{x_s \sim d_0, a_s^1 \sim \pi^s(\cdot|x_s), a_s^2 \sim \mu^s(\cdot|x_s)} \left[ \sum_{y_s = 0, 1} \left( \sqrt{\mathbb{P}_R(y = y_s | x_s, a_s^1, a_s^2)} - \sqrt{\mathcal{P}^*(y = y_s | x_s, a_s^1, a_s^2)} \right)^2 \right]$$

$$= 1 - \mathbb{E}_{x_s \sim d_0, a_s^1 \sim \pi^s(\cdot|x_s), a_s^2 \sim \mu^s(\cdot|x_s)} \left[ D_{\mathrm{H}} \left( \mathcal{P}^*(\cdot|x_s, a_s^1, a_s^2) \| \mathbb{P}_R(\cdot|x_s, a_s^1, a_s^2) \right) \right], \tag{34}$$

where $D_{\mathrm{H}}(\cdot\|\cdot)$ is Hellinger distance defined in (30). Thus by combining (33) and (34), we can derive that with probability at least $1 - \delta$, for any $R \in \mathcal{R}$, any $t \in [T]$,

$$-\frac{1}{2} \sum_{s=1}^{t-1} X_R^s \leq \sum_{s=1}^{t-1} \left[ \mathbb{E} \left[ \exp \left( -\frac{1}{2} X_R^s \right) \Big| \mathcal{F}_{s-1} \right] - 1 \right] + \log(|\mathcal{R}|/\delta)$$

$$= -\sum_{s=1}^{t-1} \mathbb{E}_{x_s \sim d_0, a_s^1 \sim \pi^s(\cdot|x_s), a_s^2 \sim \mu^s(\cdot|x_s)} D_{\mathrm{H}} \left( \mathcal{P}^*(\cdot|x_s, a_s^1, a_s^2) \| \mathbb{P}_R(\cdot|x_s, a_s^1, a_s^2) \right) + \log(|\mathcal{R}|/\delta),$$

where the first inequality comes from the fact that $\log x \leq x - 1$. Finally, by plugging in the definition of $X_R^s$, we have that with probability at least $1 - \delta$, for any $R \in \mathcal{R}$, it holds that

$$\sum_{t=1}^{T} [\mathcal{L}_{t-1}(R^*) - \mathcal{L}_{t-1}(R)] = -\sum_{t=1}^{T} \sum_{s=1}^{t-1} X_R^s$$

$$\leq -2 \sum_{t=1}^{T} \sum_{s=1}^{t-1} \mathbb{E}_{x_s \sim d_0, a_s^1 \sim \pi^s(\cdot|x_s), a_s^2 \sim \mu^s(\cdot|x_s)} D_{\mathrm{H}} \left( \mathcal{P}^*(\cdot|x_s, a_s^1, a_s^2) \| \mathbb{P}_R(\cdot|x_s, a_s^1, a_s^2) \right) + 2T \log(|\mathcal{R}|/\delta)$$

$$= -2 \sum_{t=1}^{T} \sum_{s=1}^{t-1} \mathbb{E}_{\xi_s \sim (d_0, \pi^s, \mu^s)} [\ell(R; \xi_s)] + 2T \log(|\mathcal{R}|/\delta).$$

This finishes the proof of Proposition 2. $\qquad\square$

Back to the proof of Theorem 1. We have the following decomposition of the regret,

$$\mathrm{Regret}(T)$$

$$= \sum_{t=1}^{T} \left[ V(\pi^*, \mu^*) - V(\pi^t, \dagger) \right]$$

$$= \sum_{t=1}^{T} \left[ V(\pi^*, \mu^*) - V(\pi^t, \mu^t) \right] + \sum_{t=1}^{T} \left[ V(\pi^t, \mu^t) - V(\pi^t, \dagger) \right]$$

$$= \underbrace{\sum_{t=1}^{T} \left[ V(\pi^*, \mu^*) - V_{R_1^t} \right]}_{(\heartsuit)} + \underbrace{\sum_{t=1}^{T} \left[ V_{R_1^t} - V(\pi^t, \mu^t) \right]}_{(\spadesuit)}$$

$$+ \underbrace{\sum_{t=1}^{T} \left[ V_{R_2^t}(\pi^t, \dagger) - V(\pi^t, \dagger) \right])}_{(\diamondsuit)} + \underbrace{\sum_{t=1}^{T} \left[ V(\pi^t, \mu^t) - V_{R_2^t}(\pi^t, \dagger) \right]}_{(\clubsuit)}. \tag{35}$$

We now prove the bound for $(\heartsuit, \spadesuit, \diamondsuit, \clubsuit)$ in (35).

**To bound $(\heartsuit)$.** Note that $V_{R^*} = V(\pi^*, \mu^*)$. Thus we can rewrite $(\heartsuit)$ as

$$(\heartsuit) = \sum_{t=1}^{T} \left[ V_{R^*} - V_{R_1^t} \right]. \tag{36}$$

By the choice of $R_1^t$ in (24) and Assumption 1, we have that for each $t \in [T]$,

$$V_{R^*} - \eta \cdot \mathcal{L}_{t-1}(R^*) \leq V_{R_1^t} - \eta \cdot \mathcal{L}_{t-1}(R_1^t). \tag{37}$$

By combining (36) and (37), we obtain that

$$(\heartsuit) \leq \eta \cdot \sum_{t=1}^{T} \left[ \mathcal{L}_{t-1}(R^*) - \mathcal{L}_{t-1}(R_1^t) \right]. \tag{38}$$

By Proposition 2, we can derive from (38) that with probability at least $1 - \delta$,

$$(\heartsuit) \leq -2\eta \cdot \sum_{t=1}^{T} \sum_{s=1}^{t-1} \mathbb{E}_{\xi_s \sim (d_0, \pi^s, \mu^s)} \left[ \ell(R_1^t; \xi_s) \right] + 2\eta T \log(|\mathcal{R}|/\delta). \tag{39}$$

**To bound $(\spadesuit)$.** We apply Assumption 2 and obtain that, for any $\epsilon > 0$,

$$(\spadesuit) \leq \inf_{\zeta > 0} \left\{ \frac{\zeta}{2} \sum_{t=1}^{T} \sum_{s=1}^{t-1} \mathbb{E}_{\xi_s \sim (d_0, \pi^s, \mu^s)} \left[ \ell(R_1^t; \xi_s) \right] + \frac{d(\epsilon)}{2\zeta} + \sqrt{d(\epsilon)T} + \epsilon T \right\}.$$

By taking $\zeta/2 = 2\eta$, we can further derive that

$$(\spadesuit) \leq 2\eta \cdot \sum_{t=1}^{T} \sum_{s=1}^{t-1} \mathbb{E}_{\xi_s \sim (d_0, \pi^s, \mu^s)} \left[ \ell(R_1^t; \xi_s) \right] + \frac{d(\epsilon)}{8\eta} + \sqrt{d(\epsilon)T} + \epsilon T. \tag{40}$$

**To bound ($\diamondsuit$).** Note that $V_{R^*}(\pi^t, \dagger) = V(\pi^t, \dagger)$. Thus we can rewrite ($\diamondsuit$) as

$$(\diamondsuit) = \sum_{t=1}^{T} \left[ V_{R_2^t}(\pi^t, \dagger) - V_{R^*}(\pi^t, \dagger) \right]. \tag{41}$$

By the choice of $R_2^t$ in (26) and Assumption 1, we have that for each $t \in [T]$,

$$V_{R_2^t}(\pi^t, \dagger) + \eta \cdot \mathcal{L}_{t-1}(R_2^t) \leq V_{R^*}(\pi^t, \dagger) + \eta \cdot \mathcal{L}_{t-1}(R^*). \tag{42}$$

By combining (41) and (42), we derive that

$$(\diamondsuit) \leq \eta \cdot \sum_{t=1}^{T} \left[ \mathcal{L}_{t-1}(R^*) - \mathcal{L}_{t-1}(R_2^t) \right]. \tag{43}$$

Now by Proposition 2, it further follows from (43) that with probability at least $1 - \delta$,

$$(\diamondsuit) \leq -2\eta \cdot \sum_{t=1}^{T} \sum_{s=1}^{t-1} \mathbb{E}_{\xi_s \sim (d_0, \pi^s, \mu^s)} \left[ \ell(R_2^t; \xi_s) \right] + 2\eta T \log(|\mathcal{R}|/\delta). \tag{44}$$

**To bound ($\clubsuit$).** By Assumption 2, we have that for any $\epsilon > 0$,

$$(\clubsuit) \leq \inf_{\zeta > 0} \left\{ \frac{\zeta}{2} \sum_{t=1}^{T} \sum_{s=1}^{t-1} \mathbb{E}_{\xi_s \sim (d_0, \pi^s, \mu^s)} \left[ \ell(R_2^t; \xi_s) \right] + \frac{d(\epsilon)}{2\zeta} + \sqrt{d(\epsilon)T} + \epsilon T \right\}.$$

By taking $\zeta/2 = 2\eta$, we can further derive that

$$(\clubsuit) \leq 2\eta \cdot \sum_{t=1}^{T} \sum_{s=1}^{t-1} \mathbb{E}_{\xi_s \sim (d_0, \pi^s, \mu^s)} \left[ \ell(R_2^t; \xi_s) \right] + \frac{d(\epsilon)}{8\eta} + \sqrt{d(\epsilon)T} + \epsilon T. \tag{45}$$

**Combining ($\heartsuit$), ($\spadesuit$), ($\diamondsuit$), and ($\clubsuit$).** Finally, combining (39), (40), (44) and (45), taking $\epsilon = 1/\sqrt{T}$ and

$$\eta = \frac{1}{4} \sqrt{\frac{d_{\text{TGEC}}(1/\sqrt{T})}{T \cdot \log(|\mathcal{R}|/\delta)}},$$

we can finally derive that with probability at least $1 - 2\delta$,

$$\text{Regret}(T) \leq 2 \left( \sqrt{d_{\text{TGEC}}(1/\sqrt{T}) \log(|\mathcal{R}|/\delta)} + \sqrt{d_{\text{TGEC}}(1/\sqrt{T})} + 1 \right) \sqrt{T}.$$

Thus, we finish the proof of Theorem 1. $\qquad \square$

### B.3 PROOF OF PROPOSITION 1

*Proof of Proposition 1.* This is a similar proof to that of Proposition C.11 in Liu et al. (2024a). To prove Proposition 1, we need two performance difference lemmas in the two-player zero-sum game. Given $R \in \mathcal{R}$, we define the error of $R$ with respect to the true reward $R^*$ as

$$\mathcal{E}(R; \xi) = R(x, a^1, a^2) - R^*(x, a^1, a^2), \tag{46}$$

and $\xi = (x, a^1, a^2)$. Also, we define another discrepancy function $\Delta(\cdot; \cdot)$ for theoretical analysis as

$$\Delta(R; \xi) = \left| R(x, a^1, a^2) - R^*(x, a^1, a^2) \right|^2. \tag{47}$$

**Lemma B.1** (Value decomposition for the max-player). *Let $\pi = \pi_{R_1}$ and $\mu$ be an arbitrary policy taken by the min-player. It holds that*

$$V_{R_1} - V(\pi, \mu) \leq \mathbb{E}_{\xi \sim (d_0, \pi, \mu)} \left[ \mathcal{E}(R_1; \xi) \right], \tag{48}$$

*where $\mathcal{E}(\cdot; \cdot)$ is defined in (46) and $\xi = (x, a^1, a^2)$.*

*Proof of Lemma B.1.* By the definition of $V_R$ in (20) and $V(\pi, \mu)$ in (3), we have that

$$V_{R_1} - V(\pi, \mu)$$
$$= \max_{\pi \in \Pi} \min_{\mu \in \Pi} \mathbb{E}_{d_0(x), \pi(a^1|x), \mu(a^2|x)} \left[ R_1(x, a^1, a^2) - \alpha D_{\mathrm{KL}} \left( \pi(\cdot|x) \| \pi_{\mathrm{ref}}(\cdot|x) \right) + \beta D_{\mathrm{KL}} \left( \mu(\cdot|x) \| \pi_{\mathrm{ref}}(\cdot|x) \right) \right]$$
$$\quad - \mathbb{E}_{d_0(x), \pi(a^1|x), \mu(a^2|x)} \left[ R^*(x, a^1, a^2) - \alpha D_{\mathrm{KL}} \left( \pi(\cdot|x) \| \pi_{\mathrm{ref}}(\cdot|x) \right) + \beta D_{\mathrm{KL}} \left( \mu(\cdot|x) \| \pi_{\mathrm{ref}}(\cdot|x) \right) \right]$$
$$= \min_{\mu \in \Pi} \mathbb{E}_{d_0(x), \pi_{R_1}(a^1|x), \mu(a^2|x)} \left[ R_1(x, a^1, a^2) - \alpha D_{\mathrm{KL}} \left( \pi_{R_1}(\cdot|x) \| \pi_{\mathrm{ref}}(\cdot|x) \right) + \beta D_{\mathrm{KL}} \left( \mu(\cdot|x) \| \pi_{\mathrm{ref}}(\cdot|x) \right) \right]$$
$$\quad - \mathbb{E}_{d_0(x), \pi_{R_1}(a^1|x), \mu(a^2|x)} \left[ R^*(x, a^1, a^2) - \alpha D_{\mathrm{KL}} \left( \pi_{R_1}(\cdot|x) \| \pi_{\mathrm{ref}}(\cdot|x) \right) + \beta D_{\mathrm{KL}} \left( \mu(\cdot|x) \| \pi_{\mathrm{ref}}(\cdot|x) \right) \right]$$
$$= \min_{\mu \in \Pi} \mathbb{E}_{d_0(x), \pi_{R_1}(a^1|x), \mu(a^2|x)} \left[ R_1(x, a^1, a^2) + \beta D_{\mathrm{KL}} \left( \mu(\cdot|x) \| \pi_{\mathrm{ref}}(\cdot|x) \right) \right]$$
$$\quad - \mathbb{E}_{d_0(x), \pi_{R_1}(a^1|x), \mu(a^2|x)} \left[ R^*(x, a^1, a^2) + \beta D_{\mathrm{KL}} \left( \mu(\cdot|x) \| \pi_{\mathrm{ref}}(\cdot|x) \right) \right]$$
$$\leq \mathbb{E}_{d_0(x), \pi_{R_1}(a^1|x), \mu(a^2|x)} \left[ R_1(x, a^1, a^2) + \beta D_{\mathrm{KL}} \left( \mu(\cdot|x) \| \pi_{\mathrm{ref}}(\cdot|x) \right) \right]$$
$$\quad - \mathbb{E}_{d_0(x), \pi_{R_1}(a^1|x), \mu(a^2|x)} \left[ R^*(x, a^1, a^2) + \beta D_{\mathrm{KL}} \left( \mu(\cdot|x) \| \pi_{\mathrm{ref}}(\cdot|x) \right) \right]$$
$$= \mathbb{E}_{\xi \sim (d_0, \pi, \mu)} \left[ \mathcal{E}(R_1; \xi) \right],$$

where the second equality is by the definition of $\pi_{R_1}$ in (21). $\qquad \square$

**Lemma B.2** (Value decomposition for the min-player). *Suppose that $\pi = \pi_{R_1}$ is taken by the max-player and $R_2$ is the hypothesis selected by the min-player. Let $\mu = \mu_{R_2, \pi}$ be the policy taken by the min-player. It holds that*

$$V(\pi, \mu) - V_{R_2}(\pi, \dagger) = -\mathbb{E}_{\xi \sim (d_0, \pi, \mu)} \left[ \mathcal{E}(R_2; \xi) \right], \tag{49}$$

*where $\mathcal{E}(\cdot; \cdot)$ is defined in (46) and $\xi = (x, a^1, a^2)$.*

*Proof of Lemma B.2.* By the definition of $V_R(\pi, \dagger)$ in (22) and $V(\pi, \mu)$ in (3), we have that

$$V(\pi, \mu) - V_{R_2}(\pi, \dagger)$$
$$= \mathbb{E}_{d_0(x), \pi(a^1|x), \mu(a^2|x)} \left[ R^*(x, a^1, a^2) - \alpha D_{\mathrm{KL}}(\pi(\cdot|x) \| \pi_{\mathrm{ref}}(\cdot|x)) + \beta D_{\mathrm{KL}}(\mu(\cdot|x) \| \pi_{\mathrm{ref}}(\cdot|x)) \right]$$
$$\quad - \min_{\mu \in \Pi} \mathbb{E}_{d_0(x), \pi(a^1|x), \mu(a^2|x)} \left[ R_2(x, a^1, a^2) - \alpha D_{\mathrm{KL}}(\pi_{R_1}(\cdot|x) \| \pi_{\mathrm{ref}}(\cdot|x)) + \beta D_{\mathrm{KL}}(\mu(\cdot|x) \| \pi_{\mathrm{ref}}(\cdot|x)) \right]$$
$$= \mathbb{E}_{d_0(x), \pi_{R_1}(a^1|x), \mu_{R_2, \pi_{R_1}}(a^2|x)} \left[ R^*(x, a^1, a^2) + \beta D_{\mathrm{KL}}(\mu_{R_2, \pi_{R_1}}(\cdot|x) \| \pi_{\mathrm{ref}}(\cdot|x)) \right]$$
$$\quad - \min_{\mu \in \Pi} \mathbb{E}_{d_0(x), \pi_{R_1}(a^1|x), \mu(a^2|x)} \left[ R_2(x, a^1, a^2) + \beta D_{\mathrm{KL}}(\mu(\cdot|x) \| \pi_{\mathrm{ref}}(\cdot|x)) \right]$$
$$= \mathbb{E}_{d_0(x), \pi_{R_1}(a^1|x), \mu_{R_2, \pi_{R_1}}(a^2|x)} \left[ R^*(x, a^1, a^2) + \beta D_{\mathrm{KL}}(\mu_{R_2, \pi_{R_1}}(\cdot|x) \| \pi_{\mathrm{ref}}(\cdot|x)) \right]$$
$$\quad - \mathbb{E}_{d_0(x), \pi_{R_1}(a^1|x), \mu_{R_2, \pi_{R_1}}(a^2|x)} \left[ R_2(x, a^1, a^2) + \beta D_{\mathrm{KL}}(\mu_{R_2, \pi_{R_1}}(\cdot|x) \| \pi_{\mathrm{ref}}(\cdot|x)) \right]$$
$$= -\mathbb{E}_{\xi \sim (d_0, \pi, \mu)} \left[ \mathcal{E}(R_2; \xi) \right],$$

where the third equality is by the definition of $\mu_{R, \pi}$ in (23). $\qquad \square$

Note that the right side of (49) is a general version of the right side of (48) when choosing $\mu = \mu_{R_2, \pi_{R_1}}$. Now we are ready to prove Proposition 1. Lemmas B.1 and B.2 suggest that we only need to upper-bound the term $\sum_{t=1}^{T} \left| \mathbb{E}_{\xi \sim (d_0, \pi^t, \mu^t)}[\mathcal{E}(R_2^t; \xi)] \right|$. To this end, we provide a more general result given by the following Proposition 3.

**Proposition 3.** *For a $d$-dimensional two-player zero-sum game, we assume that its expected error in (46) can be decomposed as follows*

$$\mathbb{E}_{\xi \sim (d_0, \pi^t, \mu^t)} \left[ \mathcal{E}(R_2; \xi) \right] = \langle W(R_2), X(R_2, (\pi^t, \mu^t)) \rangle, \tag{50}$$

*for some $W(R_2), X(R_2, (\pi^t, \mu^t)) \in \mathbb{R}^d$, and the discrepancy function $\Delta(R_2; \xi)$ defined in (47) can be lower bounded as follows*

$$|\langle W(R_2), X(R_2', (\pi, \mu)) \rangle|^2 \leq \mathbb{E}_{\xi \sim (d_0, \pi, \mu)}[\Delta(R_2; \xi)]. \tag{51}$$

*Also, we assume that $\|W(\cdot)\|_2 \leq B_W, \|X(\cdot, \cdot)\|_2 \leq B_X$ for some $B_W, B_X > 0$. Then it holds that*

$$\sum_{t=1}^{T} \left| \mathbb{E}_{\xi \sim (d_0, \pi^t, \mu^t)}[\mathcal{E}(R_2^t; \xi)] \right| \leq \frac{\tilde{d}(\epsilon)}{\eta} + \frac{\eta}{2} \sum_{t=1}^{T} \sum_{s=1}^{t-1} \mathbb{E}_{\xi \sim (d_0, \pi^s, \mu^s)} \left[ \Delta(R_2^t; \xi) \right]$$
$$+ \frac{\epsilon T B_W^2}{4} + 2 \min \left\{ 2 \max\{R_0, 1\} \tilde{d}(\epsilon), T \right\}$$

*for all $\epsilon \in [0, 1], \eta > 0$, and $\tilde{d}(\epsilon) := d \log(1 + T B_X^2/(d\epsilon))$.*

*Proof of Proposition 3.* This is a similar proof to that of Proposition F.3 in Liu et al. (2024a). We denote that

$$\Sigma_t = I_d + \frac{1}{\epsilon} \sum_{s=1}^{t} X(R_2^s, (\pi^s, \mu^s)) X(R_2^s, (\pi^s, \mu^s))^\top.$$

By Lemmas B.4 and B.5, we have the estimate

$$\sum_{s=1}^{t} \min \left\{ \|X(R_2^s, (\pi^s, \mu^s))\|_{\Sigma_s^{-1}}, 1 \right\} \leq 2\tilde{d}(\epsilon) \tag{52}$$

for all $\epsilon \in [0, 1]$, where $\tilde{d}(\epsilon)$ is defined in Proposition 3. Since the reward is bounded by $[0, R_0]$ by Assumption 1, we have that,

$$\sum_{t=1}^{T} \left| \mathbb{E}_{\xi \sim (d_0, \pi^t, \mu^t)}[\mathcal{E}(R_2^t; \xi)] \right|$$

$$= \sum_{t=1}^{T} \min \left\{ R_0, \langle W(R_2^t), X(R_2^t, (\pi^t, \mu^t)) \rangle \right\} \mathbf{1} \left\{ \|X(R_2^t, (\pi^t, \mu^t))\|_{\Sigma_t^{-1}} \leq 1 \right\}$$

$$+ \sum_{t=1}^{T} \min \left\{ R_0, \langle W(R_2^t), X(R_2^t, (\pi^t, \mu^t)) \rangle \right\} \mathbf{1} \left\{ \|X(R_2^t, (\pi^t, \mu^t))\|_{\Sigma_t^{-1}} > 1 \right\}$$

$$\leq \sum_{t=1}^{T} \langle W(R_2^t), X(R_2^t, (\pi^t, \mu^t)) \rangle \mathbf{1} \left\{ \|X(R_2^t, (\pi^t, \mu^t))\|_{\Sigma_t^{-1}} \leq 1 \right\} + \min\{T, 2R_0\tilde{d}(\epsilon)\}$$

$$\leq \sum_{t=1}^{T} \underbrace{\|W(R_2^t)\|_{\Sigma_t} \min \left\{ \|X(R_2^t, (\pi^t, \mu^t))\|_{\Sigma_t^{-1}}, 1 \right\}}_{(A)_t} + \min\{T, 2R_0\tilde{d}(\epsilon)\}, \tag{53}$$

where the first equality is due to the assumption in Proposition 3, the second inequality follows from (52), and the last inequality is from Cauchy-Schwarz inequality. Now we expand term $(A)_t$ in (53):

$$\|W(R_2^t)\|_{\Sigma_t} \leq \sqrt{\epsilon} B_W + \left[ \sum_{s=1}^{t-1} \left| \langle W(R_2^t), X(R_2^t, (\pi^t, \mu^t)) \rangle \right|^2 \right]^{1/2},$$

where we use $\|W(R_2^t)\|_2 \le B_W$. Then we obtain that

$$\sum_{t=1}^{T}(\mathrm{A})_t$$

$$\le \sum_{t=1}^{T}\left(\sqrt{\epsilon}B_W + \left[\sum_{s=1}^{t-1}\left|\langle W(R_2^t), X(R_2^s,(\pi^s,\mu^s))\rangle\right|^2\right]^{1/2}\right)\cdot\min\left\{\|X(R_2^t,(\pi^t,\mu^t))\|_{\Sigma_t^{-1}},1\right\}$$

$$\le \left[\sum_{t=1}^{T}\epsilon B_W^2\right]^{1/2}\cdot\left[\sum_{t=1}^{T}\min\left\{\|X(R_2^t,(\pi^t,\mu^t))\|_{\Sigma_t^{-1}},1\right\}\right]^{1/2}$$

$$+\left[\sum_{t=1}^{T}\sum_{s=1}^{t-1}\left|\langle W(R_2^t), X(R_2^t,(\pi^t,\mu^t))\rangle\right|^2\right]^{1/2}\cdot\left[\sum_{t=1}^{T}\min\left\{\|X(R_2^t,(\pi^t,\mu^t))\|_{\Sigma_t^{-1}},1\right\}\right]^{1/2}$$

$$\le \sqrt{TB_W^2\epsilon\cdot\min\{2\tilde{d}(\epsilon),T\}} + \left[2\tilde{d}(\epsilon)\sum_{t=1}^{T}\sum_{s=1}^{t-1}\left|\langle W(R_2^t), X(R_2^s,(\pi^s,\mu^s))\rangle\right|^2\right]^{1/2}$$

$$\le \sqrt{TB_W^2\epsilon\cdot\min\{2\tilde{d}(\epsilon),T\}} + \left[2\tilde{d}(\epsilon)\sum_{t=1}^{T}\sum_{s=1}^{t-1}\mathbb{E}_{\xi\sim(d_0,\pi^s,\mu^s)}[\Delta(R_2^t;\xi)]\right]^{1/2}$$

where the second inequality comes from Cauchy-Schwarz inequality, the third inequality is from (52), and the last inequality is derived from (51). Back to the analysis for (53), we have

$$\sum_{t=1}^{T}\left|\mathbb{E}_{\xi\sim(d_0,\pi^t,\mu^t)}[\mathcal{E}(R_2^t;\xi)]\right|$$

$$\le \sqrt{TB_W^2\epsilon\cdot\min\{2\tilde{d}(\epsilon),T\}} + \left[2\tilde{d}(\epsilon)\sum_{t=1}^{T}\sum_{s=1}^{t-1}\mathbb{E}_{\xi\sim(d_0,\pi^s,\mu^s)}[\Delta(R_2^t;\xi)]\right]^{1/2} + \min\{T,2R_0\tilde{d}(\epsilon)\}$$

$$\le \left[\frac{T\epsilon B_W^2}{4} + \min\{2\tilde{d}(\epsilon),T\}\right] + \left[\frac{\tilde{d}(\epsilon)}{\eta} + \frac{\eta}{2}\sum_{t=1}^{T}\sum_{s=1}^{t-1}\mathbb{E}_{\xi\sim(d_0,\pi^s,\mu^s)}[\Delta(R_2^t;\xi)]\right] + \min\{2R_0\tilde{d}(\epsilon),T\}$$

$$= \frac{\tilde{d}(\epsilon)}{\eta} + \frac{\eta}{2}\sum_{t=1}^{T}\sum_{s=1}^{t-1}\mathbb{E}_{\xi\sim(d_0,\pi^s,\mu^s)}\left[\Delta(R_2^t;\xi)\right] + \frac{\epsilon T B_W^2}{4} + 2\min\left\{2\max\{R_0,1\}\tilde{d}(\epsilon),T\right\},$$

where the second inequality is based on the AM-GM inequality, and $\eta > 0$ can be arbitrarily chosen in the last equality. Thus we finish our proof of Proposition 3. $\qquad\square$

Back to the proof of Proposition 1, we need to check the conditions of Proposition 3 for linear two-player zero-sum games. By Definition 1 and the choice of reward hypothesis class (31), we have that for any $R_2 \in \mathcal{R}$ and $\pi \in \Pi$, it holds that

$$\mathcal{E}(R_2;\xi) = R_2(x,a^1,a^2) - R^*(x,a^1,a^2) = \phi^*(x,a^1,a^2)^\top(\lambda_{R_2} - \lambda^*)$$

where $\lambda_{R_2}$ denotes the parameter of $R_2 \in \mathcal{R}$ and $\alpha^*$ is the reward parameter (see Definition 1). Thus, we can define $X(R_2,(\pi,\mu)) = \mathbb{E}_{\xi\sim(d_0,\pi,\mu)}[\phi^*(x,a,b)]$ and $W(R_2) = \lambda_{R_2} - \lambda^*$, specifying the condition (50) of Proposition 3. By Jensen's inequality and the definition of $\Delta$ in (47), one can easily see that the condition (51) of Proposition 3 holds. By the assumptions of linear two-player zero-sum games in Definition 1, we have $B_X \le 1$ and $B_W \le 2\sqrt{d}$. Thus by applying Lemma B.2

and Proposition 3, we have that

$$\sum_{t=1}^{T}\left[V(\pi^t,\mu^t)-V_{R_2^t}(\pi^t,\dagger)\right]=-\sum_{t=1}^{T}\mathbb{E}_{\xi\sim(d_0,\pi^t,\mu^t)}[\mathcal{E}(R_2^t;\xi)]$$

$$\leq\frac{\tilde{d}(\epsilon)}{\eta}+\frac{\eta}{2}\sum_{t=1}^{T}\sum_{s=1}^{t-1}\mathbb{E}_{\xi\sim(d_0,\pi^s,\mu^s)}[\Delta(R_2^t;\xi)]+\frac{T\epsilon B_W^2}{4}+2\min\left\{2\max\{R_0,1\}\tilde{d}(\epsilon),T\right\}$$

$$\leq\frac{\tilde{d}(\epsilon)}{\eta}+\frac{\eta}{2}\sum_{t=1}^{T}\sum_{s=1}^{t-1}\mathbb{E}_{\xi\sim(d_0,\pi^s,\mu^s)}[\Delta(R_2^t;\xi)]+T\epsilon d+2\sqrt{2\max\{R_0,1\}}\sqrt{\tilde{d}(\epsilon)T} \qquad (54)$$

with $\tilde{d}(\epsilon)=d\log(1+T/(d\epsilon))$ and any $\eta>0$.

Finally, we connect the discrepancy function $\Delta$ defined in (47) to the discrepancy function $\ell$ defined in (29). We denote the inverse function of the sigmoid function $\sigma$ as

$$\varsigma(z)=\log\frac{z}{1-z},z\in(0,1). \qquad (55)$$

Also, for probability distributions $P$ and $Q$ on the same probability space $(\Omega,\mathcal{F})$, the total variation distance between $P$ and $Q$ is defined as

$$D_{\mathrm{TV}}(P\|Q)=\sup_{A\in\mathcal{F}}\left\{P(A)-Q(A)\right\}. \qquad (56)$$

Notice that

$$\Delta(R;\xi)=\left|R(x,a^1,a^2)-R^*(x,a^1,a^2)\right|^2$$

$$=\left|\varsigma\left(\mathbb{P}_R(y=1|x,a^1,a^2)\right)-\varsigma\left(\mathcal{P}^*(y=1|x,a^1,a^2)\right)\right|^2$$

$$\leq\kappa^2\cdot\left|\mathbb{P}_R(y=1|x,a^1,a^2)-\mathcal{P}^*(y=1|x,a^1,a^2)\right|^2$$

$$\leq\kappa^2\cdot D_{\mathrm{TV}}\left(\mathbb{P}_R(\cdot|x,a^1,a^2)\|\mathcal{P}^*(\cdot|x,a^1,a^2)\right)^2$$

$$\leq2\kappa^2\cdot D_{\mathrm{H}}\left(\mathbb{P}_R(\cdot|x,a^1,a^2)\|\mathcal{P}^*(\cdot|x,a^1,a^2)\right)$$

$$=2\kappa^2\cdot\ell(R;\xi), \qquad (57)$$

where the second equality comes from (1), the first inequality is by Lemma B.6 with the constant $\kappa=(e^{R_0/2}+e^{-R_0/2})^2$, the second inequality is from the definition of the total variation distance in (56), and the third inequality follows from the fact that $D_{\mathrm{TV}}(P\|Q)^2\leq2D_{\mathrm{H}}(P\|Q)$. This shows that the discrepancy function defined in (29) upper-bounds the discrepancy function defined in (47) up to a factor $2\kappa^2$. Thus by plugging (57) into (54), we have

$$\sum_{t=1}^{T}\left[V(\pi^t,\mu^t)-V_{R_2^t}(\pi^t,\dagger)\right]$$

$$\leq\frac{\tilde{d}(\epsilon)}{\eta}+\eta\kappa^2\cdot\sum_{t=1}^{T}\sum_{s=1}^{t-1}\mathbb{E}_{\xi\sim(d_0,\pi^s,\mu^s)}[\ell(R_2^t;\xi)]+\epsilon Td+2\sqrt{2\max\{R_0,1\}}\sqrt{\tilde{d}(\epsilon)T}$$

$$=\frac{\bar{d}(\epsilon)}{2\eta'}+\frac{\eta'}{2}\cdot\sum_{t=1}^{T}\sum_{s=1}^{t-1}\mathbb{E}_{\xi\sim(d_0,\pi^s,\mu^s)}[\ell(R_2^t;\xi)]+\epsilon Td+\frac{\sqrt{2\max\{R_0,1\}}}{\kappa}\sqrt{\bar{d}(\epsilon)T}, \qquad (58)$$

with $\bar{d}(\epsilon)=4\kappa^2\tilde{d}(\epsilon)=4\kappa^2d\log(1+T/(d\epsilon))$ and any $\eta>0$ and $\eta'=(2\kappa^2)\eta$. This proves the second inequality of Assumption 2. For the first inequality in Assumption 2, we take $R_2^t=R_1^t,\pi=\pi_{R_1^t}$, and we can then similarly prove that

$$\sum_{t=1}^{T}\left[V_{R_1^t}-V(\pi^t,\mu^t)\right]\leq\frac{\bar{d}(\epsilon)}{2\eta'}+\frac{\eta'}{2}\cdot\sum_{t=1}^{T}\sum_{s=1}^{t-1}\mathbb{E}_{\xi\sim(d_0,\pi^s,\mu^s)}[\ell(R_1^t;\xi)]+\epsilon Td+\frac{\sqrt{2\max\{R_0,1\}}}{\kappa}\sqrt{\bar{d}(\epsilon)T}.$$

$$(59)$$

This proves that $d_{\mathrm{TGEC}}(\epsilon)\leq\bar{d}(\epsilon)$. Thus we finish the proof of Proposition 1. $\qquad\square$

### B.4 PROOF OF COROLLARY 1

*Proof of Corollary 1.* By combining (35), (39), (58), (44), (59), and Proposition 1, taking $\zeta/2 = 2\eta'$ in (58) and (59), $\epsilon = 1/\sqrt{T}$ and

$$\eta = \frac{1}{4}\sqrt{\frac{d_{\text{TGEC}}(1/\sqrt{T})}{T \cdot \log(|\mathcal{R}|/\delta)}},$$

we obtain that

$$\text{Regret}(T) \leq 2\sqrt{T}\left[4\kappa^2 d \cdot \log\left(1 + T^{2/3}/d\right)\log\left(|\mathcal{R}|/\delta\right) + 2\sqrt{2\max\{R_0, 1\}d \cdot \log\left(1 + T^{2/3}/d\right)} + d\right].$$

Thus we finish the proof of Corollary 1. $\qquad\square$

### B.5 TECHNICAL LEMMAS

**Lemma B.3** (Martingale exponential inequality)**.** *For a sequence of real-valued random variables $\{X_t\}_{t=1}^T$ adapted to a filtration $\{\mathcal{F}_t\}_{t=1}^T$, the following holds with probability at least $1 - \delta$, for any $t \in [T]$,*

$$-\sum_{s=1}^t X_s \leq \sum_{s=1}^t \log \mathbb{E}[\exp(-X_s)|\mathcal{F}_{s-1}] + \log(1/\delta).$$

*Proof of Lemma B.3.* See e.g., Theorem 13.2 (Zhang, 2023) for a detailed proof. $\qquad\square$

**Lemma B.4.** *Let $\mathcal{X} \subset \mathbb{R}^d$ and $\sup_{x \in \mathcal{X}} \|x\|_2 \leq B_X$. Then it holds that*

$$\max_{x_0, \cdots, x_{n-1} \in \mathcal{X}} \log\det\left(I_d + \frac{1}{\lambda}\sum_{t=0}^{n-1} x_t x_t^\top\right) \leq d\log\left(1 + \frac{nB_X^2}{d\lambda}\right).$$

*Proof of Lemma B.4.* See Lemma F.3. (Du et al., 2021) for a detailed proof. $\qquad\square$

**Lemma B.5** (Elliptical potential)**.** *Let $\{x_s\}_{s=1}^T$ be a sequence of vectors with $x_s \in \mathcal{V}$ for some Hilbert space $\mathcal{V}$. Let $\Lambda_0$ be a positive definite matrix and define $\Lambda_t = \Lambda_0 + \sum_{s=1}^t x_s x_s^\top$. Then it holds that*

$$\sum_{s=1}^T \min\left\{1, \|x_s\|_{\Lambda_s^{-1}}\right\} \leq 2\log\left(\frac{\det(\Lambda_T)}{\det(\Lambda_1)}\right).$$

*Proof of Lemma B.5.* See Lemma 11 (Abbasi-Yadkori et al., 2011) for a detailed proof. $\qquad\square$

**Lemma B.6** (The inverse function of sigmoid function)**.** *For any real numbers $z_1, z_2 \in [\sigma(0), \sigma(R_0)]$, it holds that*
$$|\varsigma(z_1) - \varsigma(z_2)| \leq \kappa \cdot |z_1 - z_2|,$$

where $\varsigma(z)$ is the inverse function of sigmoid function defined in (55), and the constant $\kappa = (e^{R_0/2} + e^{-R_0/2})^2$.

*Proof of Lemma B.6.* Since the function $\varsigma(\cdot)$ is differentiable on $(0, 1)$, we know that for any $z_1, z_2 \in [\sigma(0), \sigma(R_0)]$, there exists some $\rho(z_1, z_2) \in [\sigma(0), \sigma(R_0)]$, such that

$$\varsigma(z_1) - \varsigma(z_2) = \varsigma'(\rho(z_1, z_2)) \cdot (z_1 - z_2).$$

Notice that $\varsigma'(z) = \frac{1}{z(1-z)}$. We can obtain that

$$\varsigma'(\rho(z_1, z_2)) \leq \varsigma'(\sigma(R_0)) = (e^{R_0/2} + e^{-R_0/2})^2 = \kappa.$$

Thus we finish the proof of Lemma B.6. $\qquad\square$

## C    EQUIVALENCE BETWEEN MAXIMIN AND MINIMAX OBJECTIVES

In this section, we show the equivalence between the theoretical max-player target and practical max-player target under certain regularity conditions. We adopt the following assumption and theorem from Liu et al. (2024b), as they directly apply to our scenario.

First, we denote the optimization target for max-player as

$$\phi(\pi, r) \coloneqq \mathbb{E}_{x \sim d_0, a_1 \sim \pi(\cdot|x), a_2 \sim \pi(\cdot|x)} \left[ r(x, a_1) - r(x, a_2) - \alpha D_{\mathrm{KL}} \left( \pi(\cdot|x) \| \pi_{\mathrm{ref}}(\cdot|x) \right) \right] + \mathcal{L}(r), \tag{60}$$

for any $(\pi, r) \in \Pi \times \mathcal{R}$. The equivalence of maximin object and minimax object relies on the following assumption on reward function class $\mathcal{R}$.

**Assumption 4** (Regularity of reward model class (Liu et al., 2024b))**.** *We assume the reward function class $\mathcal{R}$ is a compact topological space, and the function in (60) is convex-like on $\mathcal{R}$, i.e., for any $r_1, r_2 \in \mathcal{R}$ and $\theta \in [0, 1]$, there exists $r_3 \in \mathcal{R}$ such that*

$$\phi(\pi, r_3) \leq \theta \cdot \phi(\pi, r_1) + (1 - \theta) \cdot \phi(\pi, r_2), \quad \forall \pi \in \Pi. \tag{61}$$

Based on Assumption 4, we give the equivalence between our maximin and minimax objectives in Theorem 2:

**Theorem 2** (Equivalence between maximin and minimax objectives (Liu et al., 2024b))**.** *For the policy class $\Pi$ and the reward function class $\mathcal{R}$ satisfying Assumption 4, consider the following policy defined as*

$$\pi_{\hat{r}} \in \arg \max_{\pi \in \Pi} \phi(\pi, \hat{r}), \quad where \quad \hat{r} \in \arg \min_{r \in \mathcal{R}} \max_{\pi \in \Pi} \phi(\pi, r). \tag{62}$$

*Then the policy $\pi_{\hat{r}}$ also satisfies the maximin object, i.e.,*

$$\pi_{\hat{r}} \in \arg \max_{\pi \in \Pi} \min_{r \in \mathcal{R}} \phi(\pi, r) \tag{63}$$

*Proof of Theorem 2.* See Theorem 5.6 in Liu et al. (2024b). $\square$

Theorem 2 shows that if the reward function satisfies certain conditions, the policy solving minimax problem also solves the maximin problem. This gives stronger guarantee for our practical algorithm.

## D    ADDITIONAL RESULTS ON EXPERIMENTS

Table 2 presents the results of AlpacaEval 2.0 and MT-Bench on online DPO, hybrid GSHF, SELM, SADPO, TANPO (max-player) and TANPO (min-player).

| Technique | AlpacaEval 2.0 | | MT-Bench | | |
|---|---|---|---|---|---|
| | LC Win Rate | Win Rate | Average | 1st Turn | 2nd Turn |
| Zephyr-7B-SFT (ref.) | 6.59 | 3.66 | 6.14 | 6.34 | 5.95 |
| Online DPO | 24.36 | 22.14 | 7.24 | 7.37 | 7.11 |
| Hybrid GSHF | 25.29 | 22.61 | 7.28 | 7.26 | 7.30 |
| SELM | 26.99 | 25.99 | 7.26 | 7.56 | 6.96 |
| SADPO | **28.43** | 26.21 | 7.33 | **7.71** | 6.94 |
| TANPO (max-player) | 25.05 | 23.48 | 7.24 | 7.34 | 7.13 |
| TANPO (min-player) | 27.66 | **27.08** | **7.47** | 7.55 | **7.39** |

Table 2: Full results on AlpacaEval 2.0 and MT-Bench. LC Win Rate represents Length-Controlled Win Rate.

Table 3 contains the accuracy of TANPO, SADPO and other baselines on several academic datasets. This corresponds to Figure 2 in the main text. Table 4 provides detailed win rates and length-controlled win rates of TANPO max-player and min-player across multiple iterations. These results is visualized in Figure 4 in the main text.

| Technique | GSM8k (5-shot) | MMLU | OBQA | HellaSwag (15-shot) | Winogrande (5-shot) | Average |
|---|---|---|---|---|---|---|
| Online DPO | 32.07 | 56.61 | 43.2 | 83.67 | 76.16 | 58.34 |
| SELM | 30.10 | 56.77 | **43.4** | 83.56 | 76.30 | 58.03 |
| SADPO | **33.36** | 56.96 | 43.2 | 83.40 | 76.56 | 58.70 |
| TANPO (max-palyer) | 32.75 | **57.06** | **43.4** | **83.69** | 76.40 | 58.60 |
| TANPO (min-player) | 32.84 | 56.89 | **43.4** | **83.69** | **76.64** | **58.71** |

Table 3: Results on several academic datasets.

| Metrics | Iter1 (Epoch1) | Iter2 (Epoch1) | Iter3 (Epoch1) | Iter4 (Epoch2) | Iter5 (Epoch2) | Iter6 (Epoch2) |
|---|---|---|---|---|---|---|
| TANPO (max-player) win rate | 18.45 | 20.87 | 23.48 | 23.48 | 24.72 | 25.96 |
| TANPO (max-player) LC win rate | 19.51 | 24.36 | 25.05 | 24.56 | 25.47 | 27.58 |
| TANPO (min-player) win rate | 18.45 | 21.86 | 27.08 | 26.09 | 29.55 | 29.86 |
| TANPO (min-player) LC win rate | 19.51 | 25.40 | 27.66 | 26.36 | 30.43 | 30.56 |

Table 4: TANPO (max-player) and TANPO (min-player) win rates and length-controlled (LC) win rates across 6 iterations.

# E EXPERIMENT DETAILS

## E.1 TRAINING DETAILS

We implement TANPO and SADPO along with other baselines on 4 NVIDIA A6000 GPUs. Our code is based on Alignment Handbook (Tunstall et al.). We list the training configurations in Table 5.

| | |
|---|---|
| learning rate | $5 \times 10^{-7}$ |
| learning scheduler type | cosine |
| batch size | 128 |
| warmup ratio | 0.1 |
| gradient accumulation | 16 |
| batch size per device | 2 |
| $\alpha$ | 0.01 |
| $\eta$ | 10 |
| optimizer | adamw torch |
| seed | 42 |
| precision | bfloat16 |

Table 5: Training configurations of TANPO and SADPO.

In TANPO, the response pairs are sampled from the two models using different temperature hyperparameters. We use temperature 0.7 for max-player and 0.5 for min-player. In the extended experiment for TANPO, we reduce the learning rate to $1 \times 10^{-7}$ in iteration 4 and 5, and $5 \times 10^{-8}$ in iteration 6.

## E.2 EVALUATION DETAILS

We follow the standard procedure to evaluate our model on AlpacaEval 2.0[1] and MT-Bench[2]. For AlpacaEval 2.0, we use the `alpaca_eval_gpt4_turbo_fn` as the annotators configuration, as

---

[1] https://github.com/tatsu-lab/alpaca_eval/tree/main

[2] https://github.com/lm-sys/FastChat/tree/main/fastchat/llm_judge

recommended by AlpacaEval 2.0. We use GPT-4-Turbo as the AlpacaEval 2.0 annotator. For MT-Bench, we used the default configuration, where GPT-4 is the default scoring model.

We use the default configuration in the Language Model Evaluation Harness[3] for tests on academic datasets, except in few-shot settings. For the PairRM tests, we directly input the generations from the AlpacaEval 2.0 tests into the PairRM model to evaluate the win rate.

### E.3 IMPLEMENTATION OF BASELINES

In this subsection, we discuss the details of how we implemented the baselines.

- **Online DPO.** We implement the online DPO by ourselves. Aside from the loss function, all other training setups are exactly the same as in our algorithms. It's important to note that the practical algorithm introduced by Rosset et al. (2024) is essentially equivalent to iterative DPO.

- **Hybrid GSHF.** We implement Hybrid GSHF (Xiong et al., 2023) ourselves, where two responses for each prompt are generated by the reference policy and the current policy. All other training hyperparameters are kept identical to those in our algorithms. Since Hybrid GSHF requires the reference policy to have good coverage, we use the model after one iteration of offline DPO as the reference model.

- **SELM.** We also implement SELM (Zhang et al., 2024). In Zhang et al. (2024), the training data come from the UltraFeedback dataset and generated responses. For fair comparison, we generate two responses from the policy, rank them using the same preference model, and update the policy with the SELM loss function. The training hyperparameters and response generation settings are exactly the same as those in our two-agent algorithm.

---

[3]https://github.com/EleutherAI/lm-evaluation-harness

