# OpenReview forum: "Provably Efficient and Practical Self-Play for Better LLM Alignment"
_ICLR.cc/2025/Conference — Submitted to ICLR 2025_

### Official Review · Reviewer_TBug · 2024-10-17

**Soundness:** 3
**Presentation:** 2
**Contribution:** 2
**Rating:** 5
**Confidence:** 3

**Summary:**

This work introduces a theoretical two-player zero-sum game framework for reinforcement learning from human feedback (RLHF), where the max-player approximates the Nash equilibrium and the min-player approximates the best response by each choosing their own optimistic rewards. The theoretical algorithm achieves $\tilde{O}(\sqrt{T})$ regret in linear environments. It also proposes two practical algorithms: a two-agent algorithm TANPO and a single-agent algorithm SADPO to approximate TANPO. These algorithms outperform several baseline methods in experiments.

**Strengths:**

1. This paper gives a theoretical sound result for self-play style RLHF algorithm, achieving a sub-linear regret in linear two-player zero-sum games.
2. The algorithms have better performance than baselines as shown in experiments.

**Weaknesses:**

1. The discussion on data diversity is not persuasive enough.
2. The motivation of using a single-agent approximation is not discussed.
3. As usually Nash learning results do not assume BT model or other transitive preferences, this work only focuses on BT model, lacking discussions on non-transitive preferences.
4. It would be better if the authors can plot a curve of regret to show that it is $\tilde{O} (\sqrt{T})$ (even if in toy environments).

**Questions:**

1. On top of page 4, why is $\mathcal{R}$ defined on two actions instead of one action as in Equation (3)?
2. Is the final step of Equation (7) $V_r (\pi^t, \mu)$ instead of $V_r (\pi^t, \dagger)$?
3. What is the sampling strategy of online DPO?
4. Regarding data diversity: Why does larger $|\log \pi_\text{ref} (a^1) - \log \pi_\text{ref} (a^2)|$ reflect higher data diversity? In my opinion, diversity is about the **overall distribution** of the sampled responses (more like $D_{KL} (\pi_\text{ref} (\cdot | \text{TANPO}) || \pi_\text{ref} (\cdot | \text{online DPO}))$), instead of the difference in a **local pair**.
5. Regarding single-agent approximation: What is the motivation of using a single-agent approximation? Is it more time and space efficient than two-player version?
6. Previous Nash learning results do not assume transitive preferences. Can your results applied to non-transitive preferences?

---

> ### Author Response · Authors · 2024-11-23
> **Response to Reviewer TBug (1/2)**
>
> Thank you for your constructive comment. We will address your questions point by point in the following responses. If there are any aspects we may have misunderstood or if you have further questions, we would be grateful for the opportunity to clarify and address them further.
>
> **Q1: Why is $\mathcal{R}$ defined on two actions instead of one action as in Equation (3)?**
>
> We appreciate the reviewer's question. The use of two different action spaces in defining $\mathcal{R}$ is primarily motivated by the theoretical analysis in our main theorem, which is based on the algorithm framework presented in Section 3. While our practical algorithm TANPO is equivalent to this framework when using the value function defined in Equation (3), the framework itself is more general. It is designed to apply not only to the specific value function in Equation (3) but also to any two-agent zero-sum game, including those that are not necessarily symmetric.
>
> Defining $\mathcal{R}$ with respect to two different actions allows us to accommodate this generality and ensures consistency with our theoretical proofs. However, we acknowledge that this choice might appear confusing in the current presentation. We thank the reviewer for pointing this out, and we will revise this section to make the explanation clearer and more intuitive.
>
>
> **Q2: Is the final step of Equation (7) $V_r(\pi_t,\mu)$ instead of $V_r(\pi_t,\dagger)$?**
>
> We thank the reviewer for pointing out this typo. You are absolutely correct—the final step of Equation (7) should indeed be $V_r(\pi_t,\mu)$ instead of $V_r(\pi_t,\dagger)$. We will revise this in the manuscript to ensure accuracy. We sincerely appreciate your careful review and attention to detail.
>
> **Q3: What is the sampling strategy of online DPO?**
>
> We thank the reviewer for the question. The sampling strategy in our online DPO is designed to align as closely as possible with that used in TANPO. In TANPO, to encourage more diverse response pairs, we apply different temperature values for the max-player and min-player. Specifically, we use a temperature of 0.7 for the max-player and 0.5 for the min-player. Similarly, in the online DPO setting, the responses for the two players are sampled using these respective temperatures: 0.7 for the max-player and 0.5 for the min-player. This sampling strategy is consistent with prior work, where such temperature-based methods have been shown to promote diversity in response generation [1].
>
> **Q4: Regarding data diversity: Why does larger $|\log \pi_\text{ref}(a^1) - \log \pi_\text{ref}(a^2)|$ reflect higher data diversity?**
>
> We appreciate the reviewer's insightful question regarding data diversity and its measurement. One of the motivations for exploring data diversity in our work stems from the observation that, despite the max-player in TANPO and online DPO optimizing the exact same objective, TANPO empirically outperforms online DPO. We hypothesize that this difference arises from the disparity in the training data between the two algorithms.
>
> In TANPO, the response pairs in the training data for the max-player consist of one response generated by the max-player and another generated by the min-player. In contrast, in online DPO, both responses in the pair are generated by the current policy. As a result, TANPO inherently has greater diversity within each local pair, as supported by the empirical evidence in Figure 1. To capture this, we use $|\log \pi_\text{ref}(a^1) - \log \pi_\text{ref}(a^2)|$ as a metric to quantify the diversity within local pairs. This local pair diversity directly affects the information learned by the agent and consequently impacts model performance.
>
> Furthermore, this emphasis on increasing local pair differences is consistent with exploration techniques used in RLHF literature, such as ensemble methods [2], where a central idea is to maximize the diversity between the two responses within a pair through sampling. Additionally, increasing local pair differences has been shown to be effective in practical applications, as a commonly used technique leverages a similar approach to deliberately maximize the difference between two responses in a pair through choosing large-margin pairs based on the win-rate rank [3].
>
> Regarding the reviewer's suggestion of using $D_{KL}(\pi_\text{ref}(\cdot | \text{TANPO}) || \pi_\text{ref}(\cdot | \text{online DPO}))$, we acknowledge its potential to measure overall distributional differences. However, if "TANPO" and "online DPO" in $\pi_\text{ref}(\cdot | \text{TANPO})$ and $\pi_\text{ref}(\cdot | \text{online DPO})$ refer to the prompts, the two distributions will be identical, as both TANPO and online DPO are trained on the exactly same dataset. Thus, in our work, we focus on the differences in local pairs, as they better capture the essential distinctions between the two algorithms.

---

> > ### Author Response · Authors · 2024-11-23
> > **Response to Reviewer TBug (2/2)**
> >
> > **Q5: Regarding single-agent approximation: What is the motivation of using a single-agent approximation? Is it more time and space efficient than two-player version?**
> >
> > We thank the reviewer for raising this important question. The motivation for using the single-agent approximation is to simplify the problem in certain scenarios, particularly when training efficiency is a priority. While it is true that our two-agent algorithm requires approximately twice the training time compared to the single-agent algorithm, it is important to note that the inference cost of the two-agent algorithm is lower than that of the single-agent algorithm.
> >
> > Given that, in most cases, the training cost is significantly higher than the inference cost, using the single-agent approximation can be more time and space efficient overall. This trade-off allows for more efficient training in scenarios where inference speed is a more critical concern. We hope this clarifies the rationale behind using the single-agent approximation in our work.
> >
> >
> > **Q6: Previous Nash learning results do not assume transitive preferences. Can your results applied to non-transitive preferences?**
> >
> > Thank you for your question. Our proposed algorithm framework in Section 3 is designed for a general two-agent zero-sum game and is not restricted to fully decoupled value functions. As such, it is potentially applicable to non-transitive preferences as well. However, applying the framework to non-transitive preferences may require additional approximations and modifications, which would be an interesting direction for future research. This contrasts with the BT model discussed in our paper, where the theoretical algorithm aligns seamlessly with the practical algorithm without any approximation.
> >
> > **Q7: It would be better if the authors can plot a curve of regret to show that it is $\tilde{O}(\sqrt{T})$ (even if in toy environments).**
> >
> > Thank you for the suggestion. We would like to emphasize that the $O(\sqrt{T})$ regret bound provided in our analysis is a worst-case guarantee. In practice, our algorithm may exhibit faster convergence rates depending on the specific environment and setting. Besides, similar to other RLHF literature, the number of iterations in our training process is relatively small, with only three updates per epoch. As a result, it is difficult to observe a clear regret convergence rate within this limited number of iterations. Lastly, we have provided Figure 2 in the manuscript, which shows the performance of TANPO across 6 iterations. While it does not directly plot regret, it serves as a reference for the performance progression of our algorithm.
> >
> >
> >
> > [1] Touvron H, Martin L, Stone K, et al. Llama 2: Open foundation and fine-tuned chat models[J]. arXiv preprint arXiv:2307.09288, 2023.
> >
> > [2] Dwaracherla V, Asghari S M, Hao B, et al. Efficient exploration for llms[J]. arXiv preprint arXiv:2402.00396, 2024.
> >
> > [3] Rosset C, Cheng C A, Mitra A, et al. Direct nash optimization: Teaching language models to self-improve with general preferences[J]. arXiv preprint arXiv:2404.03715, 2024.

---

> > > ### Comment · Reviewer_TBug · 2024-11-25
> > >
> > > Regarding Q6, it seems weird to me that to model a non-transitive reward requires approximation as previous papers do not assume this. Can you elaborate more on the tabular matrix game setting?
> > >
> > > Regarding Q7, I do not mean to plot a $\Theta (\sqrt{T})$ curve. It is okay if it is way better than $O(\sqrt{T})$ in some instances. For toy environments, you should be able to afford more training steps by using a small action space, say $100$ actions.

---

> > ### Comment · Reviewer_TBug · 2024-11-25
> >
> > I would like to thank the authors for their response. Regarding Q4, I basically mean for some prompt $x$, we measure the diversity of responses generated by TANPO and DPO by some divergence measure between the distribution $\pi_{TANPO} (\cdot | x)$ and $\pi_{DPO} (\cdot | x)$, where $\pi_X$ means the distribution after training with method X.

---

> ### Author Response · Authors · 2024-12-02
> **Response to Reviewer TBug**
>
> **Regarding Q4, I basically mean for some prompt $x$, we measure the diversity of responses generated by TANPO and DPO by some divergence measure between the distribution $\pi_{TANPO}(\cdot|x)$ and $\pi_{DPO}(\cdot|x)$.**
>
> We thank the reviewer for the insightful suggestion regarding the use of divergence measures between $\pi_{TANPO}(\cdot|x)$ and $\pi_{DPO}(\cdot|x)$ to assess the differences between TANPO and DPO.Our study primarily aims to understand and compare the unique characteristics of TANPO and DPO as individual algorithms, particularly focusing on why TANPO outperforms DPO. To achieve this, we use metrics that directly evaluate specific properties within each algorithm, rather than relying on divergence measures that only quantify the differences between the two. Measuring divergence alone provides limited explanatory power for understanding the underlying causes of TANPO's better performance, such as enhanced diversity or robustness.
>
> Moreover, we avoided using KL-divergence to measure the diversity within response pairs generated by DPO in the online setting because the two responses are drawn from the same policy. In such cases, KL-divergence becomes meaningless as it fails to reflect the diversity of outputs within a single distribution.
>
> **Regarding Q6, it seems weird to me that to model a non-transitive reward requires approximation as previous papers do not assume this. Can you elaborate more on the tabular matrix game setting?**
>
> We thank the reviewer for the question regarding the modeling of non-transitive rewards and the need for approximations in our algorithm. Below, we provide a detailed explanation of the tabular matrix game setting and highlight the differences from previous works.
>
> In some previous papers, such as [1][2], the proposed algorithms do not require approximations because the policy update at each step is defined as:
> $$
> \pi_{t+1} = \arg\max_{\mu} \left[\mathbb{P}(\pi \succeq \pi_t) - \alpha \, KL(\pi || \pi_t)\right],
> $$
> which has a closed-form solution:
> $$
> \pi_{t+1}(y) \propto \pi_t(y) \exp\left(\frac{\mathbb{P}(y \succeq \pi_t)}{\alpha}\right),
> $$
> and can be directly solved using standard methods. This simplicity arises because these algorithms do not explicitly account for exploration during policy updates.
>
> For the transitive reward model, we leverage a reparameterization trick to transform the exploration term in the optimization problem into a simpler and practical objective, as shown in Equation (13) of our paper. This reformulation aligns perfectly with our theoretical algorithm and does not introduce any gap. However, for the non-transitive reward model, additional approximations are needed to transform the optimization into a practical form that can be effectively applied in LLM alignment tasks. Our algorithm introduces exploration by modifying the optimization objective. Specifically, if we define $V(\pi, \mu) = \mathbb{P}(\pi \succeq \mu)$, the optimization problem for the min-player in our setting takes the following form:
> $$
> \arg\max_{\mu} \left[\hat{\mathbb{P}}(\mu \succeq \pi_t) + \eta \mathcal{L}(\mu | \mathcal{D})\right],
> $$
> where $\mathcal{L}(\mu | \mathcal{D})$ is a loss function based on historical data, and $\hat{\mathbb{P}}$ represents an estimated probability learned by the agent. Unlike the transitive setting, the exploration term $\hat{\mathbb{P}}(\mu \succeq \pi_t)$ cannot be easily reparameterized into a function of $\mu_t$, making it difficult to directly optimize. To address this, approximations may be necessary to make the optimization problem tractable and applicable in practice.
>
> In conclusion, previous works do not consider exploration in their algorithms and primarily rely on methods that avoid estimating value functions. In our case, to enable active exploration, we require an algorithm capable of effectively estimating value functions, which is inherently more challenging, especially in the non-transitive setting. We appreciate the reviewer for raising this point, as it opens up avenues for future research to improve the design of such algorithms.
>
>
> [1] Munos R, Valko M, Calandriello D, et al. Nash learning from human feedback[J]. arXiv preprint arXiv:2312.00886, 2023.
>
> [2] Wu Y, Sun Z, Yuan H, et al. Self-play preference optimization for language model alignment[J]. arXiv preprint arXiv:2405.00675, 2024.
>
> **Regarding Q7, I do not mean to plot a $\Theta(T)$ curve. It is okay if it is way better than $O(T)$ in some instances. For toy environments, you should be able to afford more training steps by using a small action space, say $100$ actions.**
>
> We thank the reviewer for the insightful suggestion. We agree that using a smaller action space could allow for more training steps, and we appreciate this direction. We will consider this approach in future work.

---

### Official Review · Reviewer_vAFV · 2024-10-19

**Soundness:** 2
**Presentation:** 2
**Contribution:** 2
**Rating:** 3
**Confidence:** 3

**Summary:**

This paper studies provably efficient self-play training algorithms of RLHF, which aims to 1) derive theoretical guarantees for the self-play framework and 2) improve with active exploration. The authors propose both an easy-to-implement two-agent algorithm and its single-agent version. Numerical results are provided to demonstrate the effectiveness of the proposed algorithms.

**Strengths:**

The proposed algorithms show improvements to the baselines in the numerical experiments.

**Weaknesses:**

1. The self-play framework is motivated by general preference models, while this paper limits to the standard BT reward model;
2. Confusion and (elementary) mathematical typos appear in critical definitions and analysis, indicating the paper's current version may not be ready for publication (see Questions below).

**Questions:**

1. Policies $\pi$ and $\mu$ are fully decoupled in the value function (3), and the minimax value is always zero, which makes the use of Nash equilibrium as a solution concept quite confusing;
2. There are typos in deriving the second equations in both (5) and (7). For instance, $\arg\max_\pi\min_\mu V_{\hat{r}_t^1}(\pi,\mu)$ in (5) is always zero while the RHS is non-zero. Additionally, there is a typo in Eq.(12).
3. A minor suggestion on writing: In Section 2, I think it would be better to introduce "Two-Agent Zero-Sum Games" in the context of the self-play training framework.

---

> ### Author Response · Authors · 2024-11-23
> **Response to Reviewer vAFV (1/2)**
>
> Thank you for your constructive comment. We will address your questions point by point in the following responses. If there are any aspects we may have misunderstood or if you have further questions, we would be grateful for the opportunity to clarify and address them further.
>
> **Q1: Policies $\pi$ and $\mu$ are fully decoupled in the value function (3).**
>
>
> We thank the reviewer for raising this insightful question regarding our choice of using a value function that can be fully decoupled while employing Nash equilibrium as a solution concept. Below, we explain our reasoning in detail:
>
> Our proposed algorithm framework in Section 3 is designed for a general two-agent zero-sum game and is not restricted to fully decoupled value functions. It is applicable to all kinds of value functions. However, for the specific case studied in this work, we chose to use a value function that can be fully decoupled for the following reasons.
>
> As demonstrated in Section 4, when using the additive value function described in Equation (3), our general algorithm framework aligns perfectly with a simple and practical algorithm. This close alignment ensures that the theoretical guarantees of the framework carry over seamlessly to the practical algorithm, which also demonstrates superior empirical performance. This highlights both the robustness and practicality of our approach in real-world scenarios.
>
> Besides, the choice of a fully decoupled value function is also motivated by our goal to find the KL-regularized optimal policy, in line with other works based on BT-model like DPO. Specifically, we still aim to solve
>    \[
>    \max_{\pi} \mathbb{E}[r(x, a)] - \alpha D_{\text{KL}}(\pi \| \pi_{\text{ref}}).
>    \]
>
> By employing a fully decoupled value function within our algorithm, achieving the Nash equilibrium directly translates to obtaining the KL-regularized optimal policy. This equivalence enables us to fairly compare our results to other works based on the BT-model.
>
> The use of Nash equilibrium as the solution is necessitated by the self-play nature of RLHF problem and our algorithm. Due to the nature of the RLHF problem, the data used for agent learning always consists of a prompt, two responses, and a preference. This structure inherently differs from standard contextual bandit problems, where a single objective can be directly maximized. While many works employ certain tricks to reformulate the RLHF problem into a contextual bandit framework, this approach often introduces practical inefficiencies. In our framework, the max-player and min-player both learn from historical data consisting of response pairs, where one response is generated by the max-player and the other by the min-player. This self-play setup inherently aligns with the Nash equilibrium concept, ensuring that both players’ strategies evolve in response to each other.
>
> Many works that model online RLHF as a single-agent optimization problem instead of a two-player zero-sum game rely on one response being generated from a fixed policy to align with the contextual bandit setting. However, in our practical implementation, both responses are generated dynamically through self-play. While theoretically, both our algorithm and single-agent optimization approaches (where one response is generated from a fixed policy) can effectively find the same optimal policy, the latter approach is impractical in real-world applications. In practice, the vast majority of algorithms adopt a self-play style, where both responses are generated dynamically from updating policies. This is also the case in our algorithm, which avoids the limitations of relying on responses generated from fixed policies. Moreover, as demonstrated empirically in Section 6, our self-play-style algorithm achieves better performance compared to single-agent approaches, further highlighting its practical advantages.
>
> We hope this explanation clarifies our design choices and demonstrates how they contribute to the robustness and effectiveness of our framework. Thank you again for the opportunity to elaborate on these points.

---

> > ### Author Response · Authors · 2024-11-23
> > **Response to Reviewer vAFV (2/2)**
> >
> > **Q2: The minimax value is always zero.**
> >
> > We thank the reviewer for raising this question. In a two-player constant-sum game, if the value function $ V(\pi, \mu) $ is symmetric with respect to its two variables $\pi$ and $\mu$, the minimax value is always a constant. This property also holds for the general preference model where $ V(\pi, \mu) = \mathbb{P}(\pi \succ \mu) $. At equilibrium, both policies $\pi$ and $\mu$ converge to the optimal policy $\pi^*$, making the minimax value $ \mathbb{P}(\pi^* \succ \pi^*) = 1/2 $[1][2].
> >
> > Furthermore, this characteristic of the minimax value provides insight into why the max-player in our framework optimizes a DPO loss. Specifically, the max-player aims to maximize the sum of the Nash equilibrium value function under the estimated reward function $\hat{r}_t^1$ and the negative estimation loss of $\hat{r}_t^1$. As shown in Equation (8), the Nash equilibrium value function under any estimated reward function is always zero, because the two inner optimization problems cancel each other out, leaving only the DPO loss.
> >
> > It is also worth noting that while the Nash equilibrium value function (i.e., the minimax value) is always zero, the best response value function is not, as it depends on the actual policy of the max-player. This distinction results in the min-player's loss function being a DPO loss with an additional exploration bonus, rather than a pure DPO loss. We believe this addresses the reviewer’s concern and clarifies the significance of the minimax value in our framework.
> >
> >
> > **Q3: There are typos in (5) and (7).**
> >
> > We sincerely apologize for the typo in the manuscript. It was an oversight on our part, and we will correcte it in the revised version. Thank you for bringing this to our attention.
> >
> > We appreciate your careful attention to the mathematical details in our manuscript. Regarding your concern about the Equation (5) involving $\arg\max_\pi$, we believe there may have been a misunderstanding about the distinction between $\arg\max$ and $\max$. The equation is indeed correct because $\arg\max_\pi$ returns the value of $\pi$ that achieves the maximum of the internal expression. While the internal expressions differ due to the addition of terms independent of $\pi$, this does not affect the resulting $\arg\max_\pi$, and hence the equality holds as stated. To clarify this point, we will add an explicit explanation in the revised manuscript. Thank you for pointing this out, as it gave us the opportunity to improve the clarity of our presentation.
> >
> > **Q4: A minor suggestion on writing: In Section 2, I think it would be better to introduce "Two-Agent Zero-Sum Games" in the context of the self-play training framework.**
> >
> > We thank the reviewer for this thoughtful suggestion regarding the presentation in Section 2. Introducing "Two-Agent Zero-Sum Games" in the context of the self-play training framework is indeed a valuable suggestion, as it can enhance the reader's understanding of the topic. We will incorporate this idea by providing a more detailed introduction to "Two-Agent Zero-Sum Games" within the context of the self-play training framework. We believe this change improves the clarity and logical flow of the section. Thank you again for this helpful input.
> >
> > **Q5: The self-play framework is motivated by general preference models, while this paper limits to the standard BT reward model.**
> >
> > Thank you for your question. Our proposed algorithm framework in Section 3 is designed for a general two-agent zero-sum game and is not restricted to fully decoupled value functions. As such, it is potentially applicable to non-transitive preferences as well. However, applying the framework to non-transitive preferences may require additional approximations and modifications, which would be an interesting direction for future research. This contrasts with the BT model discussed in our paper, where the theoretical algorithm aligns seamlessly with the practical algorithm without any approximation.
> >
> >
> > [1] Wu Y, Sun Z, Yuan H, et al. Self-play preference optimization for language model alignment[J]. arXiv preprint arXiv:2405.00675, 2024.
> >
> > [2] Munos R, Valko M, Calandriello D, et al. Nash learning from human feedback[J]. arXiv preprint arXiv:2312.00886, 2023.

---

### Official Review · Reviewer_kmze · 2024-10-31

**Soundness:** 3
**Presentation:** 3
**Contribution:** 3
**Rating:** 6
**Confidence:** 4

**Summary:**

This paper proposes a new self-play RLHF algorithm. It effectively balances the trade-off between exploration and exploitation.  In TANPO, two players are trained using different loss functions to ensure more diverse and informative data collection. And, in SADPO,  a single-agent approximation of  TANPO, which is supported by both theoretical analysis and empirical evidence. Experiments are conducted on multiple evaluation benchmarks, including AlpacaEval 2.0, MT Bench, and PairRM. It is a good paper.

**Strengths:**

This paper proposes a new self-play RLHF algorithm. It effectively balances the trade-off between exploration and exploitation.  In TANPO, two players are trained using different loss functions to ensure more diverse and informative data collection. And, in SADPO,  a single-agent approximation of  TANPO, which is supported by both theoretical analysis and empirical evidence. Experiments are conducted on multiple evaluation benchmarks, including AlpacaEval 2.0, MT Bench, and PairRM.

**Weaknesses:**

No

**Questions:**

No

---

### Official Review · Reviewer_aYgV · 2024-11-04

**Soundness:** 1
**Presentation:** 1
**Contribution:** 1
**Rating:** 3
**Confidence:** 3

**Summary:**

The submission investigates a game theoretic approach to RLHF to increase data diversity.

**Strengths:**

RLHF is an important problem.

---

Review summary: I'm not an expert in RLHF, so I'll defer to other reviewers for the strength of the submission along that axis. But as far as its game-theoretic approach, the submission seems a bit confused, as articulated in the weaknesses section. Still, RLHF is an empirical field, so the its empirical results (about which I am not qualified to speak) shouldn't be underweighted.

**Weaknesses:**

> the max-player aims to maximize the summation of (i) the expected Nash equilibrium value function and (ii) the negative estimation loss of that reward function. Similarly, the min-player seeks to maximize the summation of (i) the expected best response value function based on the max-player’s strategy and (ii) the negative estimation loss of that reward function.

These sentences doesn't make sense:
1. Value functions are already expected values.
2. Both the Nash equilibrium value function and the best response value function are a unique functions mapping from decision points to values. They are not an objective that a player can maximize.
3. The submission's use the language "that reward function", indicating a reference back to a previously mentioned reward function. But there is no such previously mentioned reward function.

---

As a aesthetic matter, "Two-Agent Nash Policy Optimization" is a pretty tasteless name. There is a large literature of policy-based algorithms for computing Nash equilibria in zero-sum games.

---

> In the Nash equilibrium, the max-player’s strategy and the min-player’s strategy are mutual best responses, meaning each is optimal given the strategy of the other (Nash et al., 1950).

-> given the other

---

> where V (π, µ) is a general function that captures the payoffs based on the strategies π and µ.

"general" is superfluous here.

---

The submission's usage of regret, while not wrong, is a bit atypical for games. Rather than quantifying a player's external regret over the iterations actually played, it quantifies that player's exploitability. Under this usage of regret, the relationship between sublinearity and convergence to Nash is less fundamental than it is games literature. Specifically, while in games literature, sublinearity is necessary for average iterate convergence, here it is just quantifying convergence speed.

---

I don't understand the motivation of setting up the equation (3) as a zero-sum game. The objective the submission describes is additively factorable across players, so you can just drop all the min terms and maximize to find the Nash equilibrium.

---

> To simplify this setup, we propose the Single-Agent Diversity driven Policy Optimization (SADPO) algorithm as a single-agent approximation of TANPO. The SADPO optimization objective is similar to min-player objective (13).

If we just care about data diversity, why bother with a game theoretic setup in the first place?

---

I don't understand Table 3. Why are there min players and max players on both axes?

**Questions:**

I included some in the weaknesses section above.

---

> ### Author Response · Authors · 2024-11-23
> **Response to Reviewer aYgV (1/3)**
>
> Thank you for your constructive comment. We will address your questions point by point in the following responses. If there are any aspects we may have misunderstood or if you have further questions, we would be grateful for the opportunity to clarify and address them further.
>
> **Q1: Value functions are already expected values.**
>
> We thank the reviewer for pointing out the redundancy in the term "expected Nash equilibrium value function." As the reviewer correctly noted, value functions in this context are already defined as expected values. We will revise the manuscript to remove this redundant phrasing and now refer to it simply as the "Nash equilibrium value function." Thank you for helping us improve the clarity of the manuscript.
>
>
> **Q2: Both the Nash equilibrium value function and the best response value function are a unique functions mapping from decision points to values. They are not an objective that a player can maximize.**
>
> We thank the reviewer for raising this important point. In our work, as well as in the majority of RLHF literature, a model-based learning approach is employed. Specifically, the max-player and min-player do not have access to the true reward function $r(x, a)$; instead, they estimate reward functions $\hat{r}_t^1$ and $\hat{r}_t^2$ based on historical data. These estimated reward functions are then used to compute the Nash equilibrium value function and the best response value function. Consequently, if the estimated reward functions differ, the resulting Nash equilibrium value function and best response value function will also differ. In this way, the agents can optimize the corresponding Nash equilibrium value function and the best response value function by changing the estimated reward function.
>
> Our optimization variables are the parameters of the estimated reward function. The objective of our framework is to find a reward function that not only minimizes the loss on the historical data but also achieves high Nash equilibrium value function and best response value function based on the estimated reward functions. This approach encourages agents to explore more effectively by guiding them towards reward functions that balance exploitation and exploration.
>
> We hope this clarifies the rationale behind our method and its objectives.
>
> **Q3: There is no such previously mentioned reward function.**
>
> We thank the reviewer for pointing out the lack of clarity regarding the reference to "that reward function." In Section 2, Equation (1), we define $r(x, a)$ as the human-provided score or a score predicted by a reward model that reflects the quality or suitability of response $a$ given the prompt $x$. However, we acknowledge that we did not explicitly refer to $r(x, a)$ as the "reward function", which might have caused confusion. While this is a common terminology in the RLHF literature, we agree that clearer terminology would improve the readability of the manuscript. To address this, we will revise the relevant sections to explicitly define $r(x, a)$ as the reward function. Thank you for your helpful suggestion.
>
> **Q4: TANPO is a tasteless name.**
>
> We appreciate the reviewer's comment on the name "Two-Agent Nash Policy Optimization." We chose this name to reflect the algorithm's focus on Nash equilibrium in two-agent zero-sum games. However, we understand that there is a large body of work on policy-based algorithms for computing Nash equilibria, and we are open to reconsidering the name to avoid any potential confusion. We will revise the manuscript accordingly. Thank you for your suggestion.
>
> **Q5: Refinements in language and presentation.**
>
> We thank the reviewer for this suggestion. We will revise the manuscript to replace "given the strategy of the other" with "given the other" as recommended. We believe this makes the statement more concise while retaining its intended meaning.
>
> We also thank the reviewer for pointing out that the term "general" is superfluous in this context. We will remove it from the manuscript to make the statement more concise. Thank you for this helpful suggestion.

---

> > ### Author Response · Authors · 2024-11-23
> > **Response to Reviewer aYgV (2/3)**
> >
> > **Q6: The objective the submission describes is additively factorable across players, so you can just drop all the min terms and maximize to find the Nash equilibrium.**
> >
> > We thank the reviewer for raising this insightful question regarding our choice of using a value function that can be fully decoupled while employing Nash equilibrium as a solution concept. Below, we explain our reasoning in detail:
> >
> > Our proposed algorithm framework in Section 3 is designed for a general two-agent zero-sum game and is not restricted to fully decoupled value functions. It is applicable to all kinds of value functions. However, for the specific case studied in this work, we chose to use a value function that can be fully decoupled for the following reasons.
> >
> > As demonstrated in Section 4, when using the additive value function described in Equation (3), our general algorithm framework aligns perfectly with a simple and practical algorithm. This close alignment ensures that the theoretical guarantees of the framework carry over seamlessly to the practical algorithm, which also demonstrates superior empirical performance. This highlights both the robustness and practicality of our approach in real-world scenarios.
> >
> > Besides, the choice of a fully decoupled value function is also motivated by our goal to find the KL-regularized optimal policy, in line with other works based on BT-model like DPO. Specifically, we still aim to solve
> >
> >    $$\max_{\pi} \mathbb{E}[r(x, a)] - \alpha D_{\text{KL}}(\pi \| \pi_{\text{ref}}).$$
> >
> >
> > By employing a fully decoupled value function within our algorithm, achieving the Nash equilibrium directly translates to obtaining the KL-regularized optimal policy. This equivalence enables us to fairly compare our results to other works based on the BT-model.
> >
> > The use of Nash equilibrium as the solution is necessitated by the self-play nature of RLHF problem and our algorithm. Due to the nature of the RLHF problem, the data used for agent learning always consists of a prompt, two responses, and a preference. This structure inherently differs from standard contextual bandit problems, where a single objective can be directly maximized. While many works employ certain tricks to reformulate the RLHF problem into a contextual bandit framework, this approach often introduces practical inefficiencies. In our framework, the max-player and min-player both learn from historical data consisting of response pairs, where one response is generated by the max-player and the other by the min-player. This self-play setup inherently aligns with the Nash equilibrium concept, ensuring that both players’ strategies evolve in response to each other.
> >
> > Many works that model online RLHF as a single-agent optimization problem instead of a two-player zero-sum game rely on one response being generated from a fixed policy to align with the contextual bandit setting. However, in our practical implementation, both responses are generated dynamically through self-play. While theoretically, both our algorithm and single-agent optimization approaches (where one response is generated from a fixed policy) can effectively find the same optimal policy, the latter approach is impractical in real-world applications. In practice, the vast majority of algorithms adopt a self-play style, where both responses are generated dynamically from updating policies. This is also the case in our algorithm, which avoids the limitations of relying on responses generated from fixed policies. Moreover, as demonstrated empirically in Section 6, our self-play-style algorithm achieves better performance compared to single-agent approaches, further highlighting its practical advantages.
> >
> > We hope this explanation clarifies our design choices and demonstrates how they contribute to the robustness and effectiveness of our framework. Thank you again for the opportunity to elaborate on these points.

---

> > > ### Author Response · Authors · 2024-11-23
> > > **Response to Reviewer aYgV (3/3)**
> > >
> > > **Q7: If we just care about data diversity, why bother with a game theoretic setup in the first place?**
> > >
> > > We appreciate the reviewer's insightful question. Our response primarily relies on empirical results. As shown in our experiments, despite the max-player using the same loss function as the baseline online DPO, the max-player achieves better empirical performance. This improvement can only be attributed to differences in the distribution of training data, as this is the sole distinction between the two approaches. We use data diversity to explain this observation and provide supporting evidence in Figure 1, where the results demonstrate the advantage of our framework in promoting data diversity.
> > >
> > > It is important to note that this perspective is not the starting point for deriving our algorithm, nor does it conflict with the game-theoretic framework. Instead, our exploration of data diversity stems from an effort to identify additional benefits provided by the framework beyond its theoretical guarantees. In RLHF, given the large and complex nature of LLM models, theoretical guarantees primarily serve as worst-case bounds and cannot fully account for superior empirical performance. In such cases, we must adopt alternative perspectives, and data diversity offers a compelling empirical explanation for the observed improvements. We hope this addresses the reviewer’s concern.
> > >
> > >
> > > **Q8: Why are there min players and max players on both axes?**
> > >
> > > We thank the reviewer for pointing out this question regarding Figure 3. In this figure, we include both the max-player and min-player from our two-agent algorithm, the LLM trained from the single-agent algorithm and the LLM trained from the online DPO as a baseline. The primary purpose of this comparison is to demonstrate that all training results from our algorithm (both the max-player and the min-player) outperform the baseline. By showing the performance of both players along with the single-agent algorithm, we provide a more comprehensive evaluation of the training outcomes. We hope this clarifies the rationale behind including both max-players and min-players on the axes in Figure 3.
> > >
> > > **Q9: The submission's usage of regret, while not wrong, is a bit atypical for games.**
> > >
> > > Thank you for pointing this out. In the field of multi-agent reinforcement learning, the usage of regret as a measure of exploitability is very common[1][2][3]. This regret reflects the extent to which the max-player can be exploited in our algorithm.
> > >
> > > [1] Jin C, Liu Q, Yu T. The power of exploiter: Provable multi-agent rl in large state spaces[C]. International Conference on Machine Learning. PMLR, 2022: 10251-10279.
> > >
> > > [2] Huang B, Lee J D, Wang Z, et al. Towards general function approximation in zero-sum markov games[J]. arXiv preprint arXiv:2107.14702, 2021.
> > >
> > > [3] Xiong W, Zhong H, Shi C, et al. A self-play posterior sampling algorithm for zero-sum markov games[C]. International Conference on Machine Learning. PMLR, 2022: 24496-24523.

---

> > ### Comment · Reviewer_aYgV · 2024-11-26
> > **Response**
> >
> > > Our optimization variables are the parameters of the estimated reward function. The objective of our framework is to find a reward function that not only minimizes the loss on the historical data but also achieves high Nash equilibrium value function and best response value function based on the estimated reward functions.
> >
> > There is the same issue here as before. These quantities "Nash equilibrium value function and best response value function" are not scalars. So it is not unambiguous to say that your objective is to make them "high".

---

> > > ### Comment · Reviewer_aYgV · 2024-11-26
> > > **Final comment**
> > >
> > > Thanks to the authors for their response. I still find myself not quite understanding the utility of a game-theoretic approach here, so I'll keep my score the same.

---

### Meta-Review · Area_Chair_VGUi · 2024-12-22

**Metareview:**

This work introduces a theoretical two-player zero-sum game framework for reinforcement learning from human feedback (RLHF), where the max-player approximates the Nash equilibrium and the min-player approximates the best response by each choosing their own optimistic rewards. The theoretical algorithm achieves sqrt-T regret in linear environments. It also proposes two practical algorithms: a two-agent algorithm TANPO and a single-agent algorithm SADPO to approximate TANPO. These algorithms outperform several baseline methods in experiments. The reviewers raised several concerns: 1) writing, 2) technical definitions, 3) limitations to BT model, 4) the motivation of using a single-agent approximation. The AC agrees with the concerns and thus recommends rejection.

**Additional Comments On Reviewer Discussion:**

The reviewers raised several concerns: 1) writing, 2) technical definitions, 3) limitations to BT model, 4) the motivation of using a single-agent approximation.  They are somewhat addressed in the rebuttal but not fully.

---

### Decision · Program_Chairs · 2025-01-22

Reject